# Noradrenaline causes a spread of association in the hippocampal cognitive map

Renée S. Koolschijn [1,2] ✉, Prakriti Parthasarathy [3], Michael Browning [4,5], Xenia Przygodda [1,6,7,8,9], Liliana P. Capitão [10], William T. Clarke [1], Tim P. Vogels [11], Jill X. O'Reilly[12,13] & Helen C. Barron[6,7,13]

The mammalian brain organises knowledge about entities in the world and relationships between them using cognitive maps. When forming a cognitive map, there is a necessary trade-off between extending the map to make novel inferences, and storing a veridical copy of past experience. However, the neural mechanisms that control this trade-off remain unknown. Using a cross-scale approach that combines a pharmacological intervention in humans with neural network modelling, we show that the neuromodulator noradrenaline elicits a significant 'spread of association' across hippocampal cognitive maps. This neural spread of association can be explained by changes in synaptic plasticity that predict overgeneralisation in behaviour. Thus, elevated noradrenaline during learning increases the 'smoothing kernel' for plasticity across the cognitive map, allowing disparate memories to become linked and distorted.

The mammalian brain stores the relationships between items and events that an organism experiences. This relational information is represented by the hippocampal system as a cognitive map of the environment[1,2]. Following its formation, a cognitive map can be used to recall information from the past, while also providing a basis to predict the consequences of upcoming actions, to guide future behaviour[3,4]. Critically, a cognitive map can extend beyond direct experience, representing more than the sum of its parts by including inferred relationships between stimuli or events in the world[5–8]. These inferred relationships can be used to draw associations that we have not directly experienced[5]. For this reason, cognitive maps are credited with enabling adaptive and flexible behaviours, such as inferential choice, generalisation and abstraction[3].

Yet, there is a fine balance to be struck when building a cognitive map. On the one hand, a cognitive map can support our ability to make highly flexible decisions, by drawing new shortcuts and inferences that extend beyond direct experience[5,9,10]. For example, Alice may supervise two PhD students, Bob and Charlie. If Bob is going to a conference, I can infer that Charlie may be going too. On the other hand, by drawing relations that deviate from reality, a cognitive map has the propensity to enable overgeneralisation by introducing potentially unwanted distortions in memory[11–13]. An example of overgeneralisation would be if I see Alice and Bob at a conference, and later 'recall' that Charlie was there as well, even though he did not in fact attend. When forming a cognitive map, the relative 'spread of association' across nodes in the cognitive map

[1]Oxford University Centre for Integrative Neuroimaging, FMRIB, Nuffield Department of Clinical Neurosciences, University of Oxford, Oxford, UK. [2]Radboud University, Donders Institute for Brain, Cognition and Behaviour, Nijmegen, the Netherlands. [3]Department of Engineering, University of Cambridge, Cambridge, UK. [4]Department of Psychiatry, University of Oxford, Warneford Hospital, Oxford, UK. [5]Oxford Health NHS Foundation Trust, Warneford Hospital, Oxford, UK. [6]Medical Research Council Brain Network Dynamics Unit, Nuffield Department of Clinical Neurosciences, University of Oxford, Oxford, UK. [7]Medical Research Council Centre of Research Excellence in Restorative Neural Dynamics, University of Oxford, Oxford, UK. [8]German Center for Neurodegenerative Diseases (DZNE), Magdeburg, Germany. [9]Institute for Cognitive Neurology and Dementia Research, Otto-von-Guericke University, Magdeburg, Germany. [10]Psychological Neuroscience Lab, Psychology Researcher Centre (CIPsi), School of Psychology, University of Minho, Campus de Gualtar, Braga, Portugal. [11]Institute of Science and Technology Austria, Klosterneuburg, Austria. [12]Department of Experimental Psychology, University of Oxford, Oxford, UK. [13]These authors contributed equally: Jill X. O'Reilly, Helen C. Barron. ✉e-mail: rskoolschijn@gmail.com

may critically determine whether the map facilitates adaptive or maladaptive behaviour.

Here we ask: what is the neural mechanism that titrates the spread of association within the cognitive map? One candidate mechanism involves the neuromodulator noradrenaline[14–16]. Noradrenaline profoundly alters plasticity in the brain by lowering the induction threshold for synaptic plasticity and reducing activity in inhibitory interneurons[17–23]. Noradrenaline may therefore titrate the 'spread of association' within the cognitive map by regulating the extent to which synaptic plasticity links discrete memories together.

To test this hypothesis, we use a synergistic cross-scale approach, applying the same behavioural paradigm to both a computational model and to human participants who undergo an fMRI scan with a double-blind pharmacological intervention. We predicted that learning during elevated noradrenaline should increase the spread of association in the underlying cognitive map.

## Results

### Experimental design

To test this prediction, in humans we manipulated noradrenaline during a learning task using a between-subject, double-blind, randomised placebo-controlled design. At the start of the experiment, participants were given either a single dose of the noradrenergic reuptake inhibitor atomoxetine ('ATX'), or a placebo ('PLC'). Previous studies demonstrate that atomoxetine can be used to increase levels of noradrenaline throughout the brain[24,25]. Our blinding procedure was effective and participants in the atomoxetine group (ATX) were unable to correctly report whether they received atomoxetine or placebo (Supplementary Fig. 1, Supplementary Table 1).

After receiving atomoxetine or placebo, participants waited 90 minutes to allow noradrenaline levels to reach peak plasma concentration[26]. Participants then completed a learning task. We designed the task such that the underlying task structure was susceptible to overgeneralisation and memory distortion. The task required participants to learn a set of pairwise associations between bird stimuli that could be arranged in a ring structure (Fig. 1a). Each bird stimulus was paired with two other bird stimuli, with each pair of birds appearing in a unique context (Fig. 1b, c, Supplementary Fig. 2). For example, bird 2 appeared both with bird 1 (in room A) and with bird 3 (in room B). Participants were not made explicitly aware of the underlying task structure, including the ring topology. Memory errors between bird stimuli and contextual cues provided a means to later test participants' tendency to overgeneralise and distort the relational task structure.

Participants learned to associate pairs of birds using a three-alternative forced choice task ('Learning task', Fig. 1d). During the learning task participants were not tested on the relationships between the bird stimuli and the contextual cues, leaving these relationships implicit. All participants learned the pairwise associations with high accuracy and there was no significant difference in learning between the ATX and PLC groups (Fig. 1e, f, Supplementary Fig. 3b).

Importantly, this task design allowed us to test whether elevated noradrenaline introduces excess inference or a 'spread of association' across the underlying cognitive map. Specifically, we predicted that if noradrenaline increases excitatory plasticity between nodes in the underlying cognitive map (Fig. 1a), stimuli and contextual cues situated in proximal locations on the ring may become erroneously associated due to a 'spread of association'. By contrast, those situated at distal locations on the ring should remain distinct, thus providing the necessary control for those in proximal locations. Our experimental design thus provided a precise and controlled means to test evidence for the effect of noradrenaline on forming a cognitive map, at both a behavioural and neural level.

### Behavioural evidence for overgeneralisation with noradrenaline

Four days after learning (Day 5, Fig. 1g), when the effects of atomoxetine had washed out[26], participants returned to perform memory tests designed to provide behavioural tests for overgeneralisation across memories. We included memory tests for both the pairwise associations between birds learned on Day 1 ('Explicit memory test', Fig. 2a; 'Ring Topology test', Supplementary Fig. 4a) and the incidental pairings between the bird stimuli and contextual cues ('Implicit memory test', Fig. 2b; 'Ring Topology test', Supplementary Fig. 4a).

For behavioural tests on Day 5 where overall accuracy in both groups was relatively high, we observed no significant group difference in explicit or implicit memory accuracy (ATX vs. PLC: Explicit memory: Fig. 2c, Supplementary Fig. 4b; Implicit memory: Fig. 2d, Supplementary Fig. 4c). Therefore, we observed no evidence to suggest that learning under elevated noradrenaline quantitatively strengthens memory encoding or consolidation of component associations. However, for the 'Implicit memory test', where overall accuracy in both groups was relatively poor, the error trials (74% of trials on average) afforded an opportunity to test evidence for overgeneralisation and distortions in memory.

Specifically, we reasoned that if the underlying memory map is distorted to include a 'spread of association' around the ring, participants should be more likely to overgeneralise by associating bird stimuli with incorrect contextual cues that are proximal rather than distal on the ring. For example, for bird 1, rather than selecting sofa A which was presented in the room where bird 1 and bird 2 were located, participants made an overgeneralisation error if they selected sofa B, which was presented in the room where bird 2 and bird 3 were located. We hypothesised that learning under noradrenaline should increase these overgeneralisation errors, due to the spread of association in the underlying cognitive map. Consistent with this prediction and regardless of the precise mathematical definition of overgeneralisation (see Methods), the ATX group compared to PLC was significantly more likely to erroneously report associations between proximal rather than distal stimuli on the ring (Fig. 2f, g; Supplementary Fig. 4d). These results suggest that learning under elevated noradrenaline increases the spread of associations between stimuli that were never presented together but are located nearby in a cognitive map.

### Physiological markers of elevated noradrenaline

In clinical trials, substantial interindividual variability is observed in response to neuropsychiatric medication, such as atomoxetine. Here, we measured this variability to quantify interindividual differences in the response to the drug in the ATX group. We used two physiological markers of noradrenergic arousal, namely the pupil dilation response and cortical measures of GABAergic tone.

### Pupil response to oddball stimuli is elevated under atomoxetine

We collected pupillometry data after the learning task during an 'oddball' detection task performed inside the MRI scanner. Work in animals and humans shows that increases in pupil diameter reflect increases in the availability of noradrenaline[28–32]. In line with the suggestion that noradrenaline signals 'unexpected uncertainty'[33], both a phasic and tonic component of the pupil dilation response are typically observed in response to an unexpected oddball stimulus[34–36]. While the phasic component involves a transient deflection from baseline, the tonic component is superimposed on the phasic component and involves a more gradual shift in baseline response that can last for several seconds to minutes[36]. The phasic component has been related to phasic bursts of locus coeruleus (LC) activity, while the tonic component has been related to baseline levels of noradrenaline[29,37] and is reported to increase with ATX[38–40].

Here, we used the pupil response to oddball stimuli as a physiological marker for noradrenaline. On each trial in the oddball detection task, participants were presented with either a familiar stimulus (91% of

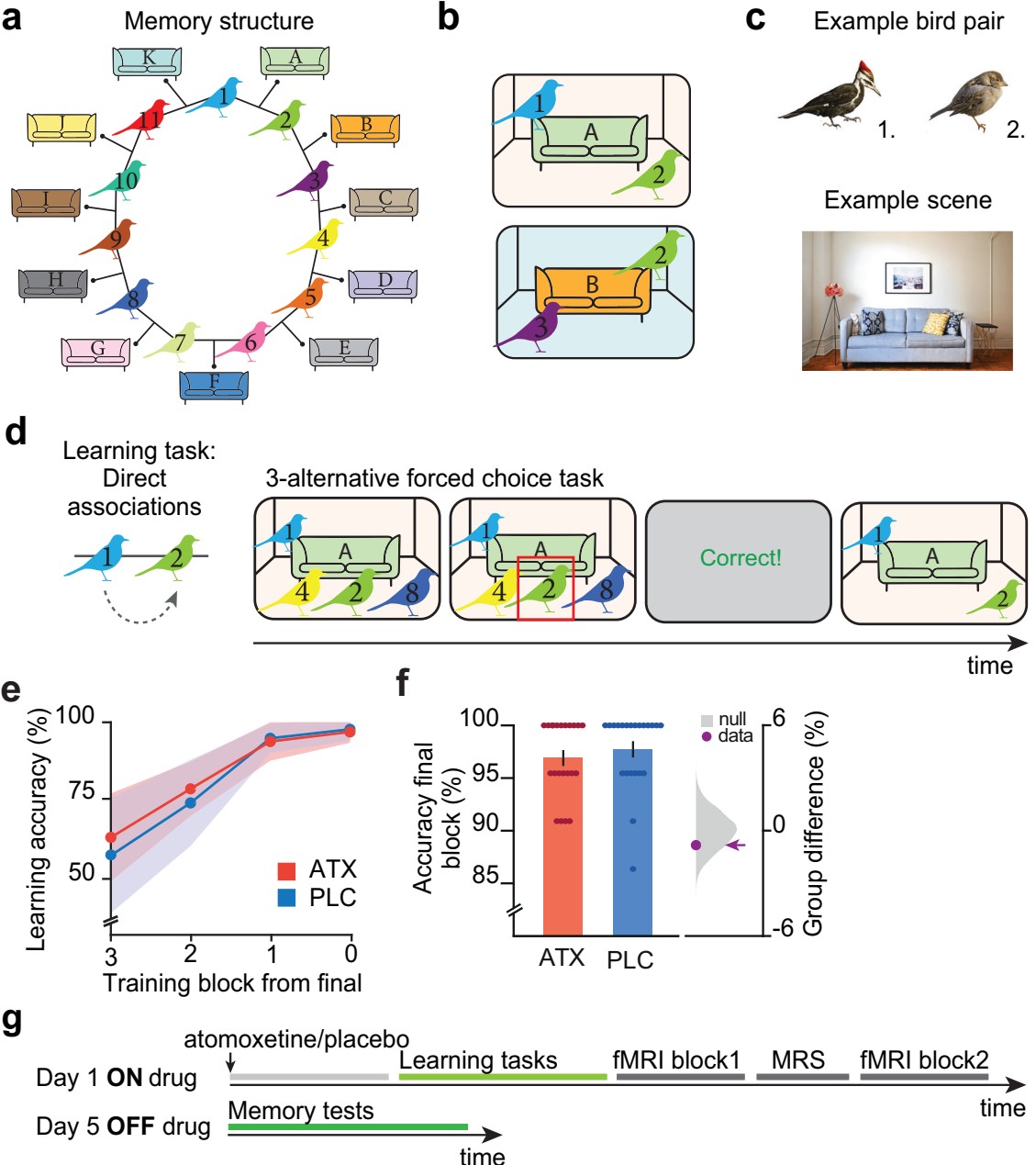

**Fig. 1 | Experimental design used to assess the effect of atomoxetine on forming a cognitive map. a**, **b** Schematic: participants explicitly learned associations between pairs of birds. Each pair of birds was shown against a particular contextual cue, with implicit exposure. Overall, each bird stimulus was associated with two other bird stimuli, such that all stimuli could be arranged as a ring structure. **c** Example pair of bird stimuli together with an example contextual cue which includes a living room scene containing a sofa. Example bird images by William Morris, example living room scene by Naomi Hébert. **d** To learn the context-dependent pairwise associations, on Day 1 participants completed a three-alternative forced choice task with feedback (Learning task). **e** Participants

performed the learning task until they successfully remembered >90% of the associations (shown: mean +/- SEM). **f** All participants successfully completed the learning task, with no significant difference between groups ((ATX − PLC ($n$ = 22:22): 2-sided permutation test: $p$ = 0.206). Left: memory accuracy (mean +/- SEM). Right: null distribution of the group differences generated by permuting subject labels, purple dot: true group difference. **g** Schematic of the experimental protocol. On Day 1, participants received either atomoxetine or placebo, completed the learning tasks and underwent an MRI scan. On Day 5, participants completed memory tests.

trials) previously observed during the learning task, or an oddball stimulus (9% of trials, Fig. 3a). Participants were required to make a button press response when they detected an oddball stimulus. We analysed pupil dilation following stimulus presentation. Consistent with previous studies[34–36,41,42], in both the ATX and PLC groups there was an enlarged phasic pupil response following oddball stimuli. The tonic component of this oddball response was greater in the ATX group compared to the PLC group, with a significant group difference

from ~6–10 s after stimulus onset (Fig. 3b). Thus, the ATX group showed an increase in their sustained tonic pupil dilation response to unexpected stimuli, providing a physiological maker for the drug response of individuals within this group.

**Elevated noradrenaline decreases inhibitory tone in neocortex**
Having established that a single dose of atomoxetine increases pupil response to surprising stimuli, we tested evidence for a second

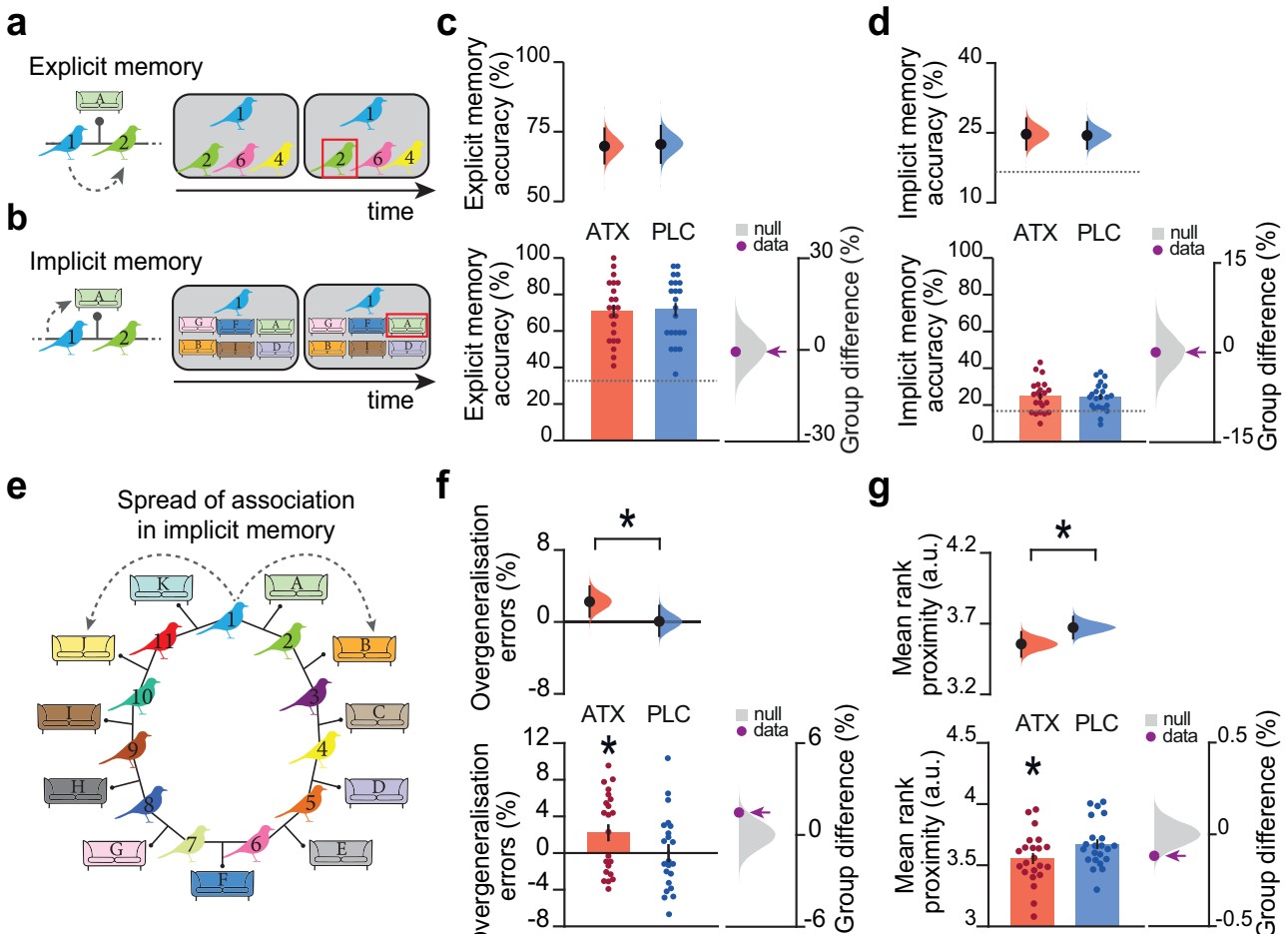

**Fig. 2 | Elevated noradrenaline during learning increases overgeneralisation in behaviour. a** Schematic of example trial in the 'Explicit memory test' performed on Day 5, where participants were tested on the bird-bird associations learned on day 1. **b** Schematic of example trial in the 'Implicit memory test' performed on Day 5, where participants were required to recall which contextual cue (sofa) was incidentally paired with each bird stimulus. **c-d, f, g** Upper: Bootstrap-coupled estimation (DABEST) plots. Black dot, mean; black ticks, 95% confidence interval; filled curve, sampling error distribution. Lower left: memory accuracy (mean +/- SEM). Lower right: null distribution of the group differences generated by permuting subject labels, purple dot: true group difference. **c** Average memory accuracy in the 'Explicit memory test' was high. There was no significant group difference in overall accuracy (ATX – PLC (*n* = 22:22): permutation test: *p* = 0.421). **d** Average memory accuracy in the 'Implicit memory test' was relatively low: on average participants made errors on 75.4% of trials. There was no significant group difference in overall accuracy (mean: ATX: 75.3%; PLC: 75.6%; ATX – PLC (*n* = 22:22): permutation test: *p* = 0.498). **e** Schematic illustrating the predicted distortion in the cognitive map, where 'spread of association' leads to overgeneralisation in participants' implicit memory. **f** 'Overgeneralisation errors' in the 'Implicit memory test' were defined as the percentage of error trials where, given the probe, the chosen contextual stimulus was neighbouring to the correct contextual stimulus. The ATX group made significantly more 'Overgeneralisation errors' than expected by chance (*p* = 0.014, effect size computed from 10,000 bias-corrected bootstrapped resamples[27]), and significantly more than the PLC group (ATX – PLC (*n* = 22:22): 2-sided permutation test *p* = 0.048). **g** 'Mean rank proximity' in the 'Implicit memory test' was defined as the average link distance between the probe and erroneously chosen contextual stimuli. For instance, a choice for sofa 'B' with probe '1' would be ranked as '2' whilst a choice for sofa 'D' would be ranked as '4'. 'Mean rank proximity' across all error trials in the implicit memory test in ATX was significantly lower than expected by chance (ATX: *n* = 21, *p* < 0.001, effect size computed from 10,000 bias-corrected bootstrapped resamples[27]) and significantly lower than in PLC (ATX – PLC (*n* = 22:22): 2-sided permutation test: *p* = 0.031).

physiological marker of elevated noradrenaline, namely, cortical inhibition. The majority of noradrenergic projections to the cortex originate in the LC, and stimulation of LC in rodents induces a decrease in cortical inhibitory tone[20]. In humans, a single dose of atomoxetine reduces paired-pulse suppression using visual evoked potentials in the visual cortex, indicating an increase in cortical excitability[43]. Taken together, we predicted that elevating noradrenaline using a single dose of atomoxetine should induce a reduction in cortical inhibition, to shift cortical activity in the direction of increased excitability.

To measure cortical inhibition in our participants, we used Magnetic Resonance Spectroscopy (MRS) (Fig. 1g) to quantify the local concentration of different neural metabolites in the brain, including the concentration of cortical gamma-aminobutyric acid (GABA+, GABA and macromolecules) together with Glx (glutamate and glutamine). We measured GABA+ from a voxel positioned in Lateral Occipital Complex (LOC) (Fig. 3c), a brain region implicated in visual object processing, including basic object recognition, scene perception and processing of object shape[44,45]. Indeed, during the scan task, the BOLD response in LOC was significantly modulated by task stimuli (Supplementary Fig. 5b). In response to ATX, any change in Glx/GABA+ in LOC should therefore result in task-relevant neural changes.

In line with previous evidence from rodents[20], GABA+ in LOC was significantly reduced in the ATX group compared to the PLC group (Fig. 3e). Moreover, this reduction in GABA+ in the ATX group accounted for a significant increase in the ratio between Glx and GABA+ in the ATX compared to the PLC group (E/I ratio, Fig. 3f)[46]. Critically, there was no significant difference in MRS quality metrics between the ATX and PLC groups (Supplementary Fig. 5c–e).

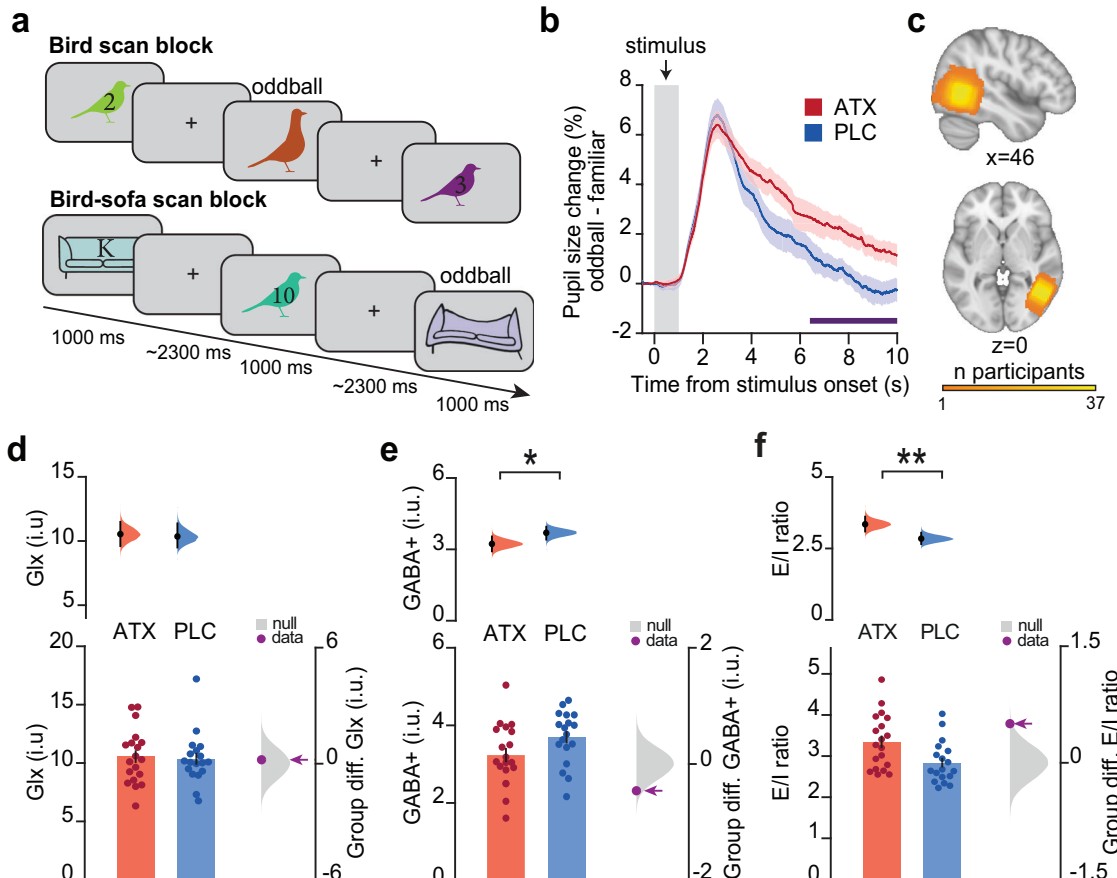

**Fig. 3 | Physiological markers of increased noradrenaline. a** During the scan task, participants observed stimuli encountered during the learning task. Participants were instructed to detect 'oddball' stimuli, encountered in ~9% of trials, which constituted warped or differently coloured versions of familiar stimuli. **b** Pupil dilation was larger in response to oddball stimuli compared to familiar stimuli, in both the ATX and PLC groups (shown: mean +/- SEM). Between 6–10 s after stimulus onset, the ATX group showed a significantly larger pupil dilation response to oddball stimuli (purple horizontal line: cluster-corrected significant ATX − PLC (*n* = 15:19); 2-sided permutation test corrected for multiple comparisons: *p* = 0.025). **c** Anatomical location of the MRS voxel, positioned in Lateral Occipital Complex (LOC). Cumulative map across *n* = 37 participants. **d**–**f** Upper:

Bootstrap-coupled estimation (DABEST) plots. Black dot, mean; black ticks, 95% confidence interval; filled curve, sampling error distribution. Lower left: Glx (**d**) and GABA+ (**e**) concentrations, and E/I ratio (**f**) (mean +/- SEM). Lower right: grey: null distribution of the group differences; purple dot: true group difference. **d** No significant group difference was observed for the concentration of Glx (ATX − PLC (*n* = 19:18): 2-sided permutation test *p* = 0.402). **e** The concentration of GABA+ was significantly lower in ATX compared to PLC (ATX − PLC (*n* = 19:18): 2-sided permutation test p = 0.034). **f** E/I ratio, defined as the ratio of Glx:GABA+ , was significantly greater in ATX compared to PLC (ATX − PLC (*n* = 19:18): 2-sided permutation test *p* = 0.007).

However, when measuring GABA+ and E/I ratio from V1, no significant difference between ATX and PLC was observed (Supplementary Fig. 6c, d). The lack of effect in V1 may be explained by lower Signal to Noise Ratio (SNR) in V1 compared to LOC, as indicated by significantly lower temporal SNR (tSNR) in this region (Supplementary Fig. 6g). In the visual object processing region LOC, significant evidence for reduced GABA+ in ATX compared to PLC suggests that a single dose of atomoxetine reduces cortical inhibition in brain regions sensitive to task stimuli. Moreover, using the ratio between Glx:GABA+ as a proxy for the balance between cortical excitation and inhibition (E/I), these results suggest that noradrenaline shifts participants' E/I ratio along a continuum toward increased cortical excitability. Taken with the pupil dilation response, this MRS measure of reduced cortical inhibition or E/I ratio in ATX provides a second physiological marker of the response to the drug.

### Reduced inhibitory plasticity during learning leads to spread of association in neural network model

Having confirmed that a single dose of atomoxetine induces expected changes in physiological markers of noradrenergic arousal, we next investigated the effect of elevated noradrenaline on the formation of a

cognitive map of the task. At a behavioural level, an increase in overgeneralisation was observed in ATX but not PLC (Fig. 2f, g). Notably, no direct relationships were observed between the physiological measures of noradrenergic arousal and these behavioural measures of overgeneralisation (Supplementary Fig. 7). We therefore asked whether physiological markers of noradrenergic arousal provide a precise predictor for neural changes in the underlying cognitive map, which in turn may predict behavioural measures of overgeneralisation.

To address this question, we first asked whether the behavioural evidence for overgeneralisation in the ATX group could be accounted for by distortions in the underlying cognitive map, i.e., the neural representation of the task. Specifically, we predicted that learning under elevated noradrenaline should introduce 'leaky' excitation around the ring structure, leading to more pronounced plasticity between proximal compared to distal nodes and establishing a spread of association in the underlying memory map. To test this hypothesis, we used a combination of computational and experimental approaches.

We built a proof-of-concept computational model to probe whether learning during a period of reduced inhibition alters synaptic modification to allow for a spread of association in the underlying

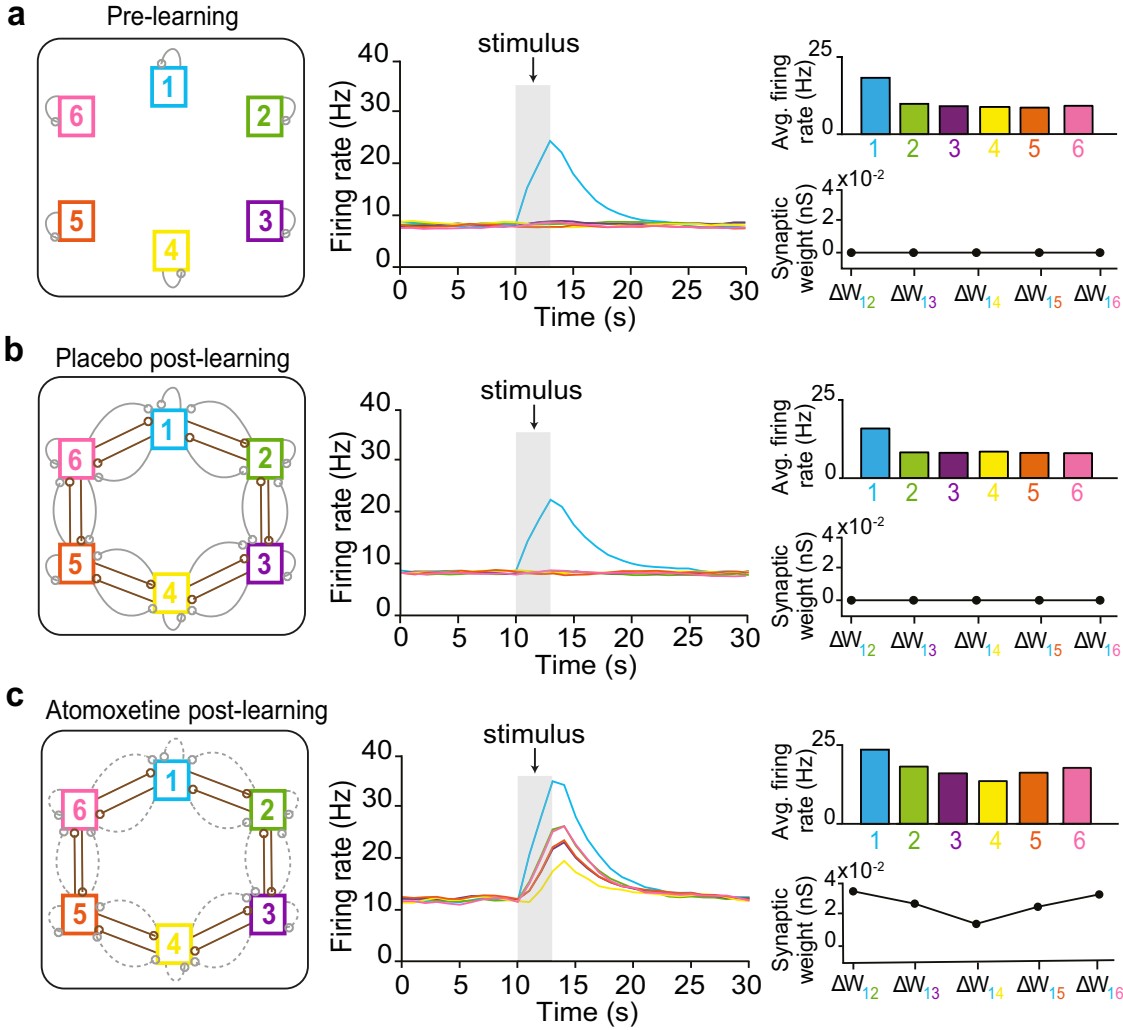

**Fig. 4 | Neural network model showing how elevated noradrenaline during learning can lead to a spread of association in the underlying memory map.** **a**–**c** Three snapshots of the recurrent spiking neural network. Left: Schematic showing the architecture and parameter conditions of the network. Six cell assemblies are pictured as coloured squares. Excitatory and inhibitory connections are drawn in brown and grey, respectively. Dotted lines indicate weaker connections. Middle: Average firing rate of all excitatory neurons in each assembly, in response to activation of assembly '1' via externally driven input ('stimulus'). Right Upper: Average firing rates of the excitatory neurons in each assembly during the 'stimulus' period. Right Lower: Change in the mean synaptic weight between assembly '1' and other assemblies following activation of assembly '1'. Additional features of the neural network are illustrated in Supplementary Fig. 8. **a** In the pre-learning state, activation of cell assembly '1' leads to high firing rates exclusively in the activated neuron group (see also Supplementary Fig. 8a). **b** Post-learning state under 'placebo', where activation of cell assembly '1' still selectively led to high firing rates in cell assembly '1' (see also Supplementary Fig. 8b). **c** Post-learning state under 'atomoxetine', where activation of cell assembly '1' also resulted in graded co-activation across neurons in the other assemblies, relative to their respective distances from assembly '1' (see also Supplementary Fig. 8c). A gradient can also be observed across the average firing rates of each assembly during the 'stimulus' period (upper right). The reduced firing in inhibitory neurons promoted a graded change in synaptic weights, embedding the spread of association in the memory map to outlast the effect of atomoxetine itself (lower right). This graded change in synaptic weights resulted in structural and functional overlap between cell assemblies. For example, assembly '1' now overlapped with each of the other assemblies by the following percentages: '2': 54%; '3': 31%. '4': 19%, '5': 28%, '6': 57%.

cognitive map. To test this hypothesis, we refined a set of previously published spiking neural network models[47–49] and utilised a co-dependent plasticity rule formalised using multiple empirical findings[47]. The co-dependent plasticity rule was applied to all excitatory synapses projecting onto excitatory cells (E-E), and all inhibitory synapses projecting onto excitatory cells (I-E). For simplicity, all remaining synapses were held static after their values were initialised, and manual changes in synaptic weight were applied to form well-defined cell-assemblies and embed the underlying ring structure.

Using our model, we replicated our experiment using a reduced ring of associations in silico. To represent distinct stimuli, we first embedded six cell assemblies, or 'nodes' 1–6, that were balanced by local intra-nodal inhibitory plasticity. To simulate the memory map in this pre-learning state (prior to learning the ring of associations), we externally activated one independent cell assembly '1' (to simulate sensory input) and found no change in the activity of other non-overlapping assemblies (Fig. 4a).

Next, we simulated the consequences of learning the ring of associations by manually embedding the underlying ring structure into the network. Excitatory (E-E) connections between neighbouring cell assemblies were strengthened, and resultant surplus excitation was balanced by strengthening relevant inhibitory (I-E) connections between neighbouring nodes, enabling clear identification of distinct excitatory and inhibitory connections associated with each assembly. Together, this resulted in the formation of a ring structure, with each node being connected to two other nodes via both excitatory and

inhibitory connections (PLC network: Fig. 4b, left). After embedding the underlying ring structure, the co-dependent plasticity rules ensured overall network stability despite external perturbations in the excitatory-inhibitory balance. Importantly, despite strong excitatory connections between neighbouring assemblies, no significant co-activation was observed along the ring structure when externally activating one cell assembly, due to the proportionally strengthened inhibitory connections (PLC network: Fig. 4b, middle-right).

In the computational model, we then simulated the effect of learning under elevated noradrenaline ('atomoxetine'). Unlike in the placebo model, in the atomoxetine model, the associations were learned under a regime of reduced global inhibitory firing across the network. Consequently, in the atomoxetine model, local inhibitory (I-E) connections between nodes were only minimally strengthened in response to learning (ATX network: Fig. 4c, left). This resulted in an excitatory surplus that was not balanced by synaptic inhibition, leading to elevated excitatory plasticity. These changes manifested as a gradient of co-activation along the ring structure in response to activation of a single cell assembly '1' (ATX network: Fig. 4c, middle-right). The graded co-activation occurred relative to their respective distances from assembly '1', with greater co-activation of nearby assemblies ('2' and '6') in contrast to more distal assemblies ('3' and '5', and '4'). This graded co-activation further led to graded plasticity across cell assemblies, indicated by the graded changes in synaptic weights between pairs of cell assemblies (Fig. 4c, middle-right). Thus, in our proof-of-principle model, learning under reduced inhibitory firing was sufficient to embed a spread of association in the underlying memory map.

Using our neural network model, we further explored alternative mechanisms that can account for a spread of association in the underlying memory map. Consistent with previous modelling of associative memory maps[50], in the post-learning PLC network we show that a reduction in local (weakening only inter-nodal I-E connections) but not global (weakening all network E-I connections) inhibition is sufficient to reveal a spread of association (Supplementary Fig. 9). This reduction in local inhibition may plausibly occur via transient disinhibition during memory recall[51,52]. However, we note that when memory recall performance was tested on Day 5 (Fig. 2; Supplementary Fig. 4), the effect of the drug had washed out, giving no reason to expect a group difference in local inhibition at this time point.

Another possibility is that the spread of association in the ATX group is due to increased assembly overlap. Indeed, some degree of both functional and structural assembly overlap is expected across neighbouring assemblies due to associative learning of bird-bird pairs, in line with empirical evidence[53,54] and computational models[50,55,56]. In the context of our model, we explored how assembly overlaps influence overgeneralisation. In the PLC network, we found that overgeneralisation is observed when neighbouring assemblies overlap by ≥40%, a value in part determined by network size (Supplementary Fig. 10). However, given that no significant evidence for overgeneralisation was observed in the PLC group (Fig. 2f, g; Supplementary Fig. 4d), we conclude that assembly overlap due to mere associative learning is not sufficient to drive overgeneralisation behaviour. Rather, given that we observe significant overgeneralisation in the ATX group, and a significant group difference (ATX – PLC) (Fig. 2f, g; Supplementary Fig. 4d), this suggests that elevated noradrenaline during learning increases cell assembly overlap. Indeed, this is precisely what we observe in Fig. 4c, where graded co-activity across the network results in graded synaptic plasticity, increasing functional overlap between cell assemblies in a graded manner. Overall, our computational model suggests that when learning occurs during a period of reduced inhibition, local inhibitory rebalancing is compromised, leading to synaptic modifications that embed a spread of association across memories in the underlying cognitive map.

## Elevated noradrenaline increases spread of association in the underlying cognitive map

To empirically test the predictions of our model in our participants (Figs. 1–3), we used functional Magnetic Resonance Imaging (fMRI) to measure neural representations of task stimuli and representations of the underlying memory map (Fig. 1g). Previous studies in both animals and humans demonstrate that the hippocampus represents both spatial and abstract cognitive maps of the environment[1,3]. Thus, in the ATX group, we predicted hippocampal representations that reflect distortions in the overall task structure. In addition, based on our use of an object-in-scene (bird-in-room) paradigm, we predicted neural representations of task cues in visual object and scene processing regions, namely the LOC and the parahippocampal cortex. Moreover, the parahippocampal cortex provides a major input-output route for information processing between the sensory cortex (e.g., LOC) and the hippocampal formation[57,58] (Fig. 5a). Analyses were therefore performed across these three regions of interest: LOC, parahippocampus and hippocampus, together with the whole imaged brain volume.

To measure neural representations using fMRI we took advantage of repetition suppression, which relies on the fact that neurons show a relative suppression in their activity when presented with repeated stimuli to which they are responsive[59,60]. While typically used with fMRI to access sub-voxel representations for single stimuli[61], 'cross-stimulus' suppression can be used to index the relative co-activation or overlap between neural representations coding for two different stimuli[48]. To measure cross-stimulus suppression, stimuli were presented in a pseudo-randomised order during the fMRI scan task (Fig. 3a). Each stimulus was categorised post hoc according to the node distance of the stimulus that came immediately before, as defined by the underlying ring structure (Fig. 5b). For example, bird stimuli preceded by an associated bird were categorised as proximal '1 node' trials (e.g. bird '1' preceded by bird '2'), while bird stimuli preceded by birds further away on the ring structure were categorised accordingly (e.g. '4 nodes': bird '1' preceded by bird '5') (Fig. 5b). Notably, the symmetrical ring topology of the underlying memory map provided an efficient way to measure node distance from each node.

We reasoned that if noradrenaline leads to a spread of association in the underlying memory map, we should observe greater co-activity, and therefore greater cross-stimulus suppression, for proximal compared to distal nodes. To test this prediction, we first identified brain regions where cross-stimulus suppression scales across all possible node distances. Consistent with our prediction, in the ATX group we observed significant evidence for a spread of association across the underlying memory map in right hippocampus and parahippocampus (Fig. 5c, Supplementary Table 3). No significant effect was observed in the PLC group (Supplementary Fig. 12b). While no significant group differences between ATX and PLC were observed across the hippocampus-parahippocampus region of interest (Fig. 5e), in parahippocampus there was a trend towards a significant group difference (Fig. 5f). These results show qualitative agreement between our data and neural network model.

We next sought to *quantitatively* compare our data to the output of the neural network model. Notably, as our neural network model was simplified to include only 6 cell assemblies (Fig. 4), we sought to identify brain regions where cross-stimulus suppression scales with node distance as quantified using the simulated firing rates generated by our neural network model in response to stimulating assembly '1' (ATX model: Fig. 4c). Thus, for each node (Fig. 1a) we assessed evidence for a spread of association up to a maximum distance of 3 nodes apart. In the ATX group we again observed significant evidence for a spread of association across the underlying memory map in the right hippocampus (Fig. 5g, Supplementary Fig. 12c, Supplementary Table 4). No significant effect was observed in the PLC group (Supplementary Fig. 12d). While no significant group differences between ATX and PLC were observed across the hippocampus-parahippocampus region of

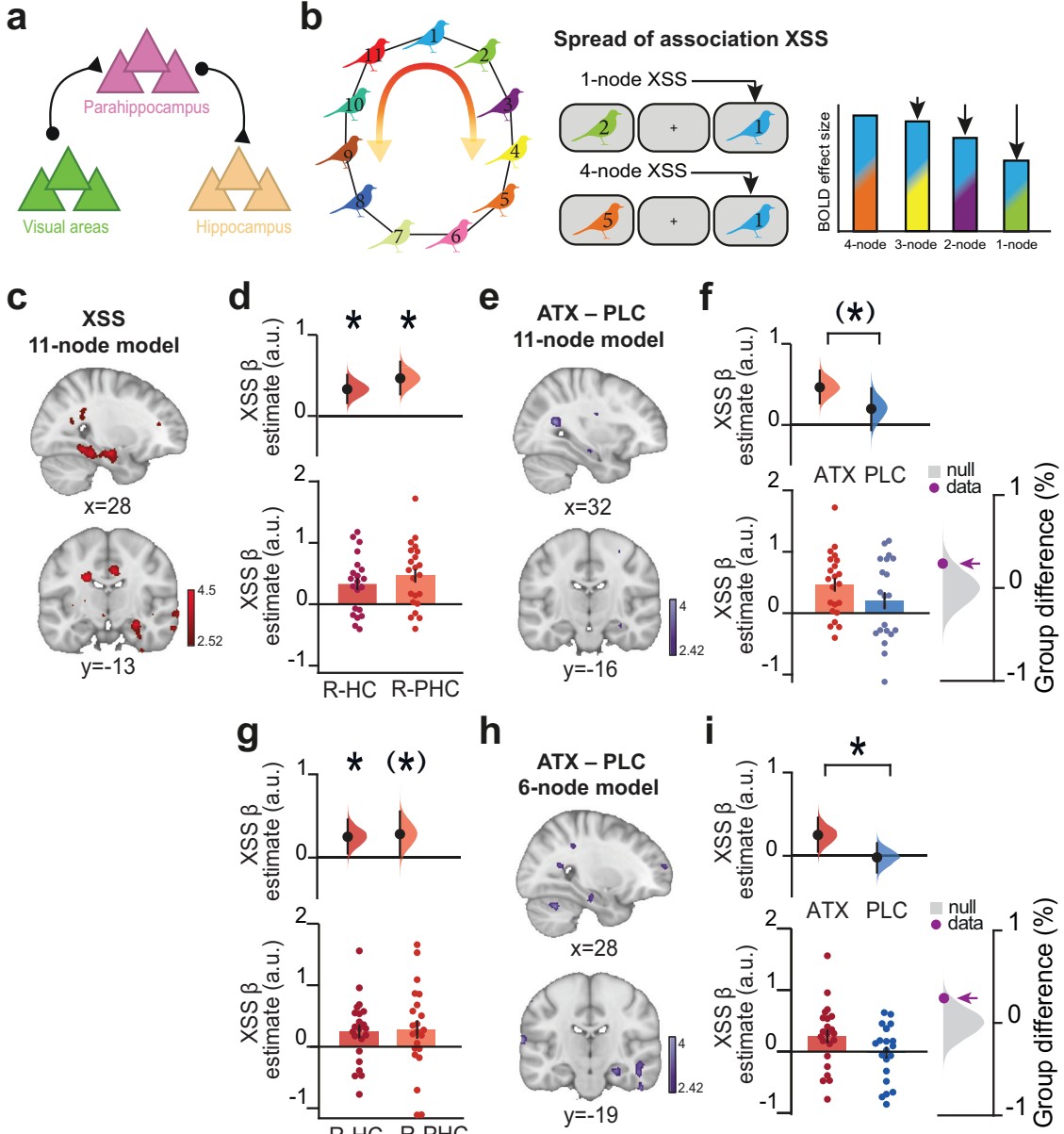

**Fig. 5 | Elevated noradrenaline induces a neural spread of association in the hippocampus and parahippocampus. a** Schematic illustrating the hypothesised regions involved in representing the memory map (Fig. 1a). **b** Schematic of the cross-stimulus suppression (XSS) fMRI contrast. **c-e** Evidence for neural spread of association in ATX. **c** T-statistic map showing XSS for spread of association in ATX, with a significant effect in right hippocampus and right parahippocampal cortex (SVC with parahippocampus-hippocampus ROI: $n = 22$, $t_{21} = 4.56$, $p = 0.032$, MNI coordinates, Supplementary Table 3). Post hoc tests revealed significant effects in right parahippocampus ($n = 22$, $t_{21} = 4.56$, $p = 0.005$), left parahippocampus ($n = 22$, $t_{21} = 3.51$, $p = 0.017$), and right hippocampus ($n = 22$, $t_{21} = 4.35$, $p = 0.017$). **d**, **f**, **g**, **i** Lower: individual β parameter estimates (mean +/- SEM). Upper: DABEST plots. Black dot, mean; black ticks, 95% confidence interval; filled curve, sampling error distribution. **d** In ATX, a significant spread of association effect was observed in right hippocampus (R-HC: $n = 22$, $p = 0.003$, 1-sided effect size computed from 10,000 bias-corrected bootstrapped resamples[27]) and right parahippocampus (R-PHC: $n = 22$, $p = 0.0012$, effect size computed from 10,000 bias-

corrected bootstrapped resamples[27]). **e** T-statistic map showing no significant group differences for the spread of association effect (SVC with parahippocampus-hippocampus ROI, ATX − PLC ($n = 22$:$21$): $t_{41} = 2.35$, $p = 0.595$, MNI coordinates). **f** Post hoc tests revealed a trend group difference in right parahippocampus for the spread of association effect (ATX − PLC ($n = 22$:$21$): $p = 0.041$, 2-sided permutation test). **g–i** Quantitative comparison between the ATX neural network model (Fig. 4c) and data. **g** In ATX, a significant spread of association effect was observed in right hippocampus (ATX: $n = 22$, $p = 0.024$), with a trend in right parahippocampus (ATX: $n = 22$, $p = 0.055$) 1-sided effect sizes computed from 10,000 bias-corrected bootstrapped resamples[27] (mean +/- SEM). **h** T-statistic map showing a trend group difference for the spread of association effect (SVC with parahippocampus-hippocampus ROI: ATX − PLC ($n = 22$:$21$), $t_{41} = 2.96$, $p = 0.136$, MNI coordinates, Supplementary Table 5). **i** Post hoc tests revealed a significant group difference in right hippocampus for the spread of association effect (ATX − PLC ($n = 22$:$21$): $p = 0.041$, 2-sided permutation test).

interest (Fig. 5h, Supplementary Table 5), a post hoc test revealed a significant group difference in right hippocampus (Fig. 5h, i). Taken together, these results support the predictions from the model by suggesting that learning under elevated noradrenaline leads to a spread of association in the underlying cognitive map.

As the response to the drug was variable in the ATX group, in subsequent analyses of the underlying neural representations we asked whether this variance could be explained by behavioural measures and our two physiological markers indicating individual differences in the response to the drug.

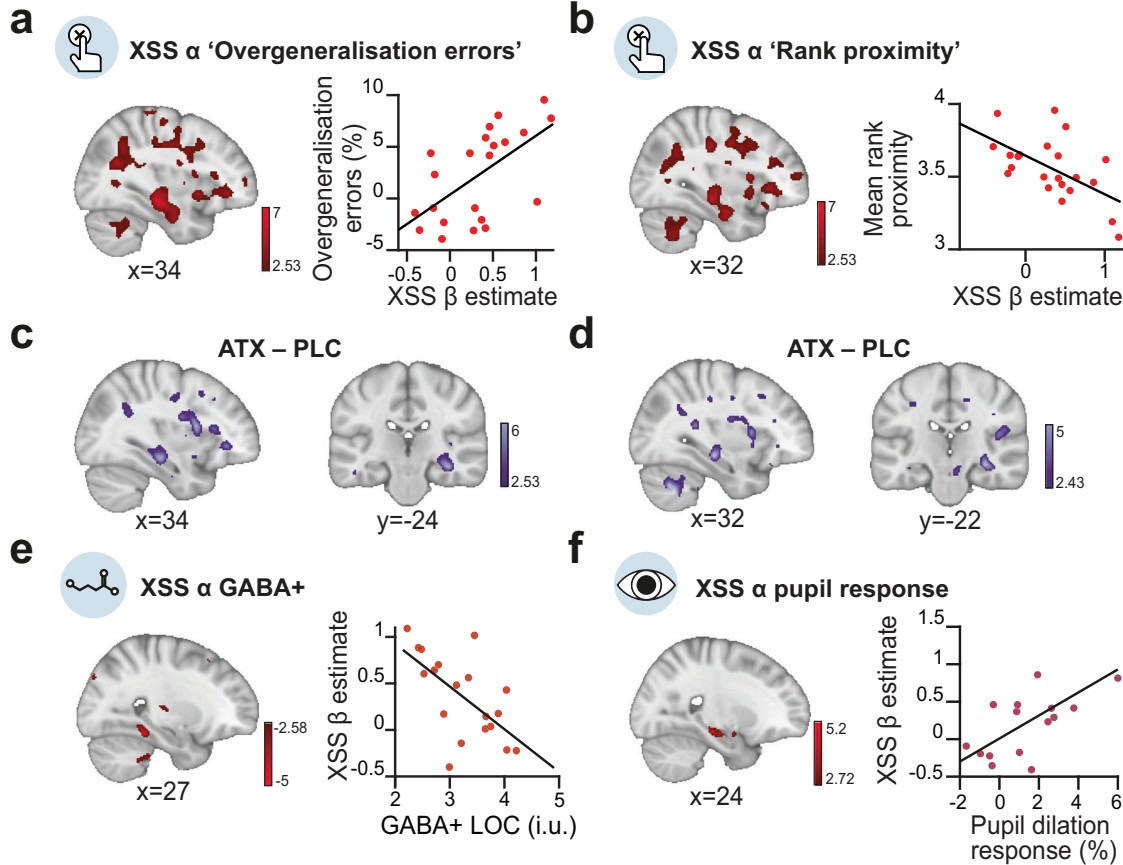

**Fig. 6 | Neural spread of association predicts behavioural measures of overgeneralisation and is predicted by physiological measures of reduced inhibitory tone. a** Left: In ATX, neural spread of association in hippocampus and parahippocampal cortex predicted 'Overgeneralisation errors' (SVC with parahippocampus-hippocampus ROI, n = 22, $t_{20}$ = 6.16, $p$ = 0.002, MNI coordinates, Supplementary Table 6). Post hoc tests revealed a significant effect in right hippocampus (SVC with right hippocampus ROI, $n$ = 22, $t_{20}$ = 6.16, $p$ = 0.001). Right: correlation plot using extracted parameter estimates from right hippocampus (Supplementary Fig. 12a) (Spearman correlation: n = 22, $r_{20}$ = 0.680, $p$ > 0.001). **b** Left: In ATX, neural spread of association in hippocampus and parahippocampal cortex predicted 'Mean rank proximity' (SVC with parahippocampus-hippocampus ROI, n = 22, $t_{20}$ = 6.16, p = 0.002, MNI coordinates, Supplementary Table 7). Post hoc tests revealed a significant effect in right hippocampus (SVC with right hippocampus ROI, n = 22, $t_{20}$ = 5.92, p = 0.001). Right: correlation plot using extracted parameter estimates from right hippocampus (Spearman correlation: n = 22, $r_{20}$ = −0.595, p = 0.004). **c** Group difference for the correlation between neural spread of association and 'Overgeneralisation errors' (ATX − PLC (n = 22:21): SVC with parahippocampus-hippocampus ROI, $t_{41}$ = 4.51, p = 0.011, MNI coordinates, Supplementary Table 8). Post hoc tests revealed a significant effect in right hippocampus (ATX − PLC (n = 22:21): SVC with right hippocampus ROI, $t_{41}$ = 4.51, p = 0.004). **d** Group difference for the correlation between the neural spread of

association and 'Mean proximity rank' (ATX − PLC, SVC with parahippocampus-hippocampus ROI, $t_{41}$ = 4.51, $p$ = 0.011, MNI coordinates, Supplementary Table 9). Post hoc tests revealed a significant effect in right hippocampus (ATX − PLC (n = 22:21): SVC with right hippocampus ROI, $t_{41}$ = 4.03, $p$ = 0.013). **e** Left: In ATX, GABA+ in LOC predicted neural spread of association in hippocampus and parahippocampal cortex (SVC with parahippocampus-hippocampus ROI, n = 19, $t_{16}$ = 4.70, p = 0.047, MNI coordinates, Supplementary Table 10). Post hoc tests revealed a significant correlation in right and left parahippocampus (SVC with left parahippocampus ROI, n = 19, $t_{16}$ = 4.70, $p$ = 0.008; SVC with right parahippocampus ROI, $n$ = 19, $t_{16}$ = 4.07, $p$ = 0.023). Right: Correlation plot using extracted parameter estimates in right parahippocampus (Spearman correlation: $n$ = 19, $r_{17}$ = −0.661, $p$ = 0.003). **f** Left: In ATX, we observed a positive trend between pupil dilation response to surprising stimuli (Fig. 3b) and neural spread of association (SVC with parahippocampus-hippocampus ROI, n = 15, $t_{11}$ = 5.15, $p$ = 0.066, MNI coordinates, Supplementary Table 11). Post hoc tests revealed a significant correlation in right hippocampus (SVC with right hippocampus ROI, $n$ = 15, $t_{11}$ = 5.15, $p$ = 0.023). Right: correlation plot using extracted parameter estimates from right hippocampus (Spearman correlation: $n$ = 15, $r_{13}$ = 0.511, $p$ = 0.054). T-statistic maps are thresholded at $p$ < 0.01 uncorrected, for visualisation purposes.

First, we asked whether the neural spread of association in the underlying memory map observed immediately after learning (Fig. 5d, g) can predict overgeneralisation in behaviour on day 5 (Fig. 2f, g). We reasoned that participants who show stronger neural spread of association in their underlying memory map immediately after learning should retain this knowledge and show stronger evidence for spread of association in subsequent behaviour. In the ATX group, we observed a significant positive relationship between the neural measures of spread of association in hippocampus and parahippocampal cortex and overgeneralisation errors reported in behaviour using two different behavioural measures (Fig. 6a-b). The significant peak of the effect was observed in hippocampus, verified

using post hoc tests (Supplementary Table 6, Supplementary Table 7). Therefore, participants who showed a greater spread of association in the underlying memory map on Day 1 went on to show a greater spread of association in their behaviour on Day 5. Notably, no significant effects were observed for the PLC group (Supplementary Fig. 13a,b) and a significant difference was observed between the ATX and PLC groups, again using two different behavioural measures (Fig. 6c-d, Supplementary Table 8, Supplementary Table 9). These results suggest that learning under elevated noradrenaline leads to a spread of association in the underlying cognitive map, manifesting as systematic and sustained overgeneralisation in the behavioural readout of memory.

### Physiological markers of noradrenaline predict neural spread of association

These findings are observed after only a single dose of atomoxetine with variable effects across individuals. Importantly, here we measured the individual response to atomoxetine using two physiological markers of noradrenergic arousal, namely the significant increase in pupil dilation response to surprising stimuli (Fig. 3b) and the significant reduction in GABAergic tone in LOC (Fig. 3e). Using these two physiological markers as a measure for the response to atomoxetine, in the ATX group we asked whether individual differences in the neural spread of association in parahippocampus-hippocampus measured using fMRI can be predicted by the physiological response to the drug.

Within the ATX group, we observed a significant negative relationship between GABA+ in LOC and the neural spread of association in parahippocampus-hippocampus (Fig. 6e). The peak of the effect was observed in parahippocampus, verified using post hoc tests (Supplementary Table 10). Notably, there was no significant evidence for this relationship in the PLC group (Supplementary Fig. 13c), though no significant group difference was observed (Supplementary Fig. 13 d). Therefore, participants in the ATX group who learned the memory map with lower GABA+ levels in LOC showed stronger neural spread of associations in their underlying memory map. This suggests that the spread of association in the underlying memory map can be attributed to learning in a state of reduced inhibitory tone, which differentially affects plasticity between proximal compared to distal nodes on the ring.

We next asked whether our second physiological measure of response to the drug, namely the pupil response to surprising stimuli, could also predict the neural spread of association in parahippocampus-hippocampus. Within the ATX group, we observed a positive trend between the pupil dilation response and the neural spread of association in parahippocampus-hippocampus (Fig. 6f). The peak of the effect was significant in hippocampus, verified using post hoc tests (Supplementary Table 11). Therefore, participants in the ATX group who showed a stronger pupil response also showed a stronger neural spread of association in their representation of the underlying memory map in the hippocampus. No significant relationship was observed in the PLC group (Supplementary Fig. 13e) and significant group differences were observed in both hippocampus and parahippocampus, verified using post hoc tests (Supplementary Fig. 13f, Supplementary Table 12). Taken together, these results suggest that the significant neural spread of association in the underlying memory map of the ATX group can be predicted by the variance in the response to atomoxetine. Moreover, a significant between-group difference is observed for this relationship when assessing variance in the drug response, as captured by pupil dilation.

## Discussion

When forming a cognitive map, there is a necessary trade-off between extending the map to make novel inferences versus storing a veridical copy of past experience. Here, using a double-blind placebo-controlled pharmacological intervention paradigm, we show that the neuromodulator noradrenaline titrates this trade-off. Specifically, we show that learning under elevated noradrenaline increases the neural spread of association in the underlying cognitive map. This neural spread of association predicts overgeneralisation errors in behaviour that persist even after the effect of the pharmacological intervention has washed out.

To provide a mechanistic explanation for these findings, we use both empirical data and a proof-of-concept spiking neural network model. We show that when noradrenaline is elevated, the concentration of cortical GABA+ is reduced, consistent with previous studies in animal models[20]. We use a spiking neural network model to explore the consequences of learning a cognitive map under reduced inhibitory tone. We show that when learning occurs during a window of reduced inhibitory tone, graded co-activity and synaptic plasticity occur

between nodes in a cognitive map. This graded synaptic plasticity in the model can account for the significant neural spread of association we observe in the parahippocampus and hippocampus. Critically, this neural spread of association can be predicted by the reduction in GABA+ and other measures of noradrenergic arousal, namely the pupil dilation response to surprising stimuli. Moreover, the neural spread of association can predict behavioural measures of overgeneralisation acquired days later, after the effect of atomoxetine has washed out. These results suggest that learning under elevated noradrenaline reduces cortical inhibition, thereby spreading association around a cognitive map via graded synaptic plasticity. In this manner, noradrenaline may be considered to set the size of the 'smoothing kernel' applied to nodes within a cognitive map.

Previous studies show that noradrenaline modulates the induction threshold for synaptic plasticity[17–19,22,23] by dramatically reducing firing of inhibitory interneurons in cortex[20]. Our model demonstrates how learning map-like representations under a global reduction in inhibition embeds a spread of association across memories that outlasts the initial plasticity. Specifically, elevated noradrenaline reduces global inhibition to allow for a regime of enhanced excitatory synaptic plasticity[62] and reduced inhibitory plasticity[63]. Inhibitory plasticity is thought to play a critical role in forming 'inhibitory engrams'[64] to stabilise new learning[48,65] and prevent memory interference[66]. Our findings show how learning during a period of elevated noradrenaline can prevent the formation of effective inhibitory engrams, with lasting consequences for behaviour that are underpinned by changes in the organisation of memory.

Noradrenaline may therefore enable the formation of cognitive maps that promote predictions that go beyond direct experience, to allow agents to take novel shortcuts[2,6,8], infer unobserved relationships[5], and generalise information across discrete experiences[67]. By entering a regime of enhanced synaptic plasticity, learning under elevated noradrenaline may promote representation of unobserved associations, by 'joining-the-dots' between nearby nodes in the cognitive map. This mechanism may complement activity observed in the hippocampus during periods of rest and sleep[68–70], where spiking sequences are generated that extend beyond direct experience[5,9].

Noradrenaline has been associated with signalling salience and arousal by modulating attention and neural gain[37,39,71]. In the extreme, learning in states of high noradrenergic arousal may render a memory maladaptive, as observed in Post-Traumatic Stress Disorder (PTSD), where levels of noradrenaline are higher than healthy controls and predict the severity of clinical symptoms[72,73]. Our findings demonstrate *how* memory distortions and systematic biases in memory can be introduced, raising an important prediction that learning under elevated noradrenaline can account for clinically significant maladaptive memories in PTSD. By setting the width of the 'smoothing kernel' for plasticity, noradrenergic tone during learning may control the extent to which memories become linked within a cognitive map, ultimately parametrising the organisation of knowledge.

To measure participants' response to atomoxetine, we used two physiological measures: MRS, which provided an index of cortical excitability, and pupil dilation to surprising stimuli. In the ATX compared to the PLC group we observed significant changes in these physiological measures. Moreover, these physiological measures provided a precise predictor of the neural and, in turn, behavioural measures of overgeneralisation. Notably, we observed no significant direct relationship between these physiological measures and behavioural measures of overgeneralisation errors. Thus, a measure of the neural response to atomoxetine is necessary to provide a reliable predictor of overgeneralisation errors in behaviour. These findings mirror previous research demonstrating that cortical excitability, measured using MRS, provides a precise prediction for neural and, in turn, behavioural measures of memory interference[66].

Together, our data and neural network model provide a testbed for various competing hypotheses that explain overgeneralisation behaviours. First, we show no significant difference in overall memory accuracy between ATX and PLC on both explicit and implicit memory tests, suggesting no significant quantitative difference in the strength of memory encoding or consolidation of component associations. This suggests that overgeneralisation behaviour in the ATX group cannot be explained by enhanced representation of the entire ring topology. Instead, our neural data suggests that overgeneralisation behaviour in the ATX group is best explained by the effect of noradrenaline on graded co-activity in the underlying representation of the ring topology (or cognitive map).

Importantly, our results show that elevated noradrenaline during learning affects not only the co-activation of stimulus representations in the immediate post-learning phase, but also the level of generalisation in the cognitive map five days later when the drug has completely washed out. Our proof-of-concept neural network model suggests a mechanism by which this can occur: overgeneralisation can be explained by 'leaky' excitation in the underlying memory map, which allows graded co-activity between assemblies. This 'leaky' excitation manifests when inhibitory connections fail to counterbalance excitatory co-activity. This failure in inhibitory counterbalance may occur at different time points relative to learning. We show that both reducing inhibition during learning and local disinhibition after learning promote graded co-activity between assemblies, where the latter may be considered to model transient local disinhibition that accompanies memory recall[46,51,52]. However, here we observe a significant group difference in overgeneralisation between ATX and PLC on Day 5, after the effect of the drug has washed out and when no local group difference in local disinhibition is expected. Instead, our data imply differential encoding of the underlying cognitive map, where dialling inhibition down during learning (due to elevated noradrenaline) embeds a cognitive map with 'leaky' excitation across memories.

Our neural network model also provides an opportunity to test the effect of assembly overlap on overgeneralisation. Previous computational models elegantly demonstrate that assembly overlap can in part account for spread of association in an underlying memory map[50]. Moreover, empirical evidence suggests that associative learning increases assembly overlap[53,54]. To explore the relationship between functional assembly overlap and overgeneralisation, in our PLC neural network model we varied the degree of assembly overlap. We show that overgeneralisation is observed only in the PLC network when neighbouring assemblies overlap by ≥40%, a value that is partly determined by the network size. However, given that our PLC group shows no significant evidence for overgeneralisation errors, we conclude that assembly overlap due to mere associative learning is not sufficient to drive overgeneralisation behaviour. Rather, given we observe a significant group difference in overgeneralisation between ATX and PLC, we infer that elevated noradrenaline during learning increases cell assembly overlap. Indeed, in our ATX model, we observe a graded increase in cell assembly overlap due to excess excitatory synaptic plasticity during learning. We note that the effect of noradrenaline on cell assembly overlap may depend on the type of memory in question and vary across brain regions. For example, evidence in rodents suggests that elevated noradrenaline in basal-lateral amygdala but not central amygdala or hippocampus can increase specificity of a contextual fear memory[74]. This variation across brain regions may be attributed to differences in the local composition and diversity of inhibitory interneurons. Overall, our proof-of-principle neural network model provides traction on the precise cellular and synaptic mechanisms that underlie the significant spread of association we observe in the cognitive map during learning under elevated noradrenaline.

## Methods

### Participants and double blinding

44 healthy volunteers participated in the study. We implemented a double-blind, randomised, placebo-controlled study design. Participants were randomly assigned to one of two groups, which were stratified by sex: one group that received a single dose of 10 mg atomoxetine (ATX group, $n = 22$, mean age: $23.9 +/- 5.23$ yrs, 11 women) and one group who received placebo (PLC group, $n = 22$, mean age: $25.2 +/- 4.80$ yrs, 11 women). Participants reported no history of psychiatric disease, and were not taking any prescription medicines within the 2 weeks prior to the time of testing, with the exception of the oral contraceptive pill. All experiments were approved by the University of Oxford ethics committee (reference number R60579/RE004). All participants gave informed written consent.

Randomisation was performed by a researcher not involved in recruitment and data collection (LC). The drugs were administered by an experimenter (RK), who was blind to the drug condition. At the end of the final testing day, participants and experimenter indicated whether they thought the participant had received atomoxetine or a placebo.

The effectiveness of the double blinding procedure was assessed using the Bang's Blinding Index (BI)[75]. A score of 1 indicates that all responses were correct and complete unblinding is inferred; a score of −1 indicates that all responses were incorrect and complete blinding is inferred or unblinding in the opposite direction (e.g., opposite guessing); a score of 0 indicates that half the guesses were correct and half of the guesses were incorrect, inferring successful blinding with random guessing. Scores between −0.2 and 0.2 indicate that blinding was successful.

### Behavioural training

On the first day of the experiment, all participants ($n = 44$) were given an over-encapsulated capsule, which contained either a single dose of 10 mg atomoxetine (ATX group) or an empty capsule (PLC group). Participants were then required to wait for 90 min (Fig. 1g) to allow time for the drug to take effect[26].

Following the waiting period, participants learned 11 pairwise associations between bird stimuli. Each pair of birds was presented in a particular context, where the context constituted a living room interior (Fig. 1b,c). The bird stimuli were always presented in the same location within each particular context (Supplementary Fig. 2). Together, the context dependent pairwise associations formed a symmetrical ring structure (Fig. 1a). This ring structure provided an efficient way to later measure representational overlap between proximal and distal associative links as each bird could be used as a control for an association on the other side of the ring (see *fMRI scan task*). Participants were not made aware of the underlying ring structure.

To learn the bird-bird associations, participants performed a 3-alternative forced choice task with feedback ('Learning task', Fig. 1d). In every trial, participants were shown a background stimulus with one of the bird stimuli in place. Choice stimuli could be displayed upon pressing the buttons 'b','n', or 'm' on the keyboard and participants confirmed their choice by pressing one of these three buttons together with the spacebar. There was no response deadline. Feedback was subsequently presented for 1.5 s, after which the background with the correct stimulus pairing was shown until the participant again pressed spacebar to proceed to the next trial. Participants were required to score >90% correct on the 3-alternative forced choice learning task for two consecutive blocks. Upon reaching this criterion, participants completed the 'Inference task' (Supplementary Fig. 3a), which involved participants indicating which bird stimulus shared an association with the probe bird (e.g., probe bird: bird '1'; inferred choice: bird '3') thus making trajectories along the ring structure. The inference task was performed in the absence of contextual cues or feedback, again using a

3-alternative forced choice test. Importantly, throughout these training tasks, the bird-bird associations were made explicit while the bird-sofa relationships remained implicit.

Four days later, on Day 5, participants completed a battery of tests to assess memory for explicitly and implicitly learned associations. The battery of tests included the following: (1) 'Explicit Memory Test', where a 3-alternative forced choice task was used to test memory for bird-bird associations in the absence of contextual cues or feedback (Fig. 2a); (2) a repeat of the 'Inference Test', as encountered on Day 1 and described above (Supplementary Fig. 3a); (3) 'Implicit Memory Test', where a 6-alternative forced choice task was used to test memory for bird-context associations in the absence of feedback (Fig. 2b); (4) 'Ring Topology Test', where participants were given the opportunity to construct the ring topology by placing each bird pairing in the correct scene using drag-and-drop (Supplementary Fig. 4a), completed after all other tests. The 'Ring topology test' was collected in all participants apart from the first, resulting in $n = 43$ for this task.

In the Implicit Memory Test, images of sofas taken from the background interiors were used as contextual cues, rather than the full image of the living room interior. Importantly, throughout the task, the bird stimuli were always presented in the same location within their contextual interior but never on or proximal to the sofas.

For each test, the average accuracy was estimated for each participant. On the 'Implicit Memory Test' (Fig. 2b), as overall memory accuracy was close to chance (Fig. 2d), we analysed the error trials to estimate two behavioural measures for the relative spread of association between stimuli. The first measure sought to quantify 'Overgeneralisation errors', defined as:

$$
\begin{aligned}
&Overgeneralisation\,errors \\
&= \frac{\sum_{trial=1}^{ntrials}\left(\frac{proximal\,choice_{trial}}{N\,proximal\,cues_{trial}}\right) - \sum_{trial=1}^{ntrials}\left(\frac{distal\,choice_{trial}}{N\,distal\,cues_{trial}}\right)}{N\,trials\,with\,opportunity\,for\,overgeneralisation\,error} \times 100\%
\end{aligned}
\tag{1}
$$

i.e., the percentage of trials where participants erroneously linked a bird stimulus to a neighbouring ('proximal') contextual cue in the underlying ring structure (Fig. 2e, e.g. selecting sofa 'B' when prompted with bird '1'), minus the percentage of trials where any other 'distal' error was made (e.g. selecting sofa 'E' when prompted with bird '1'), divided by the total number of trials where there was the opportunity for overgeneralisation errors. The second measure sought to measure 'Mean rank proximity', defined as: the average distance in links between the probe and the erroneously chosen contextual cue. For instance, given probe '1', a choice for sofa 'B' would be ranked as '2' whilst a choice for sofa 'D' would be ranked as '4'. Participants with a high 'Overgeneralisation error' score and a low 'Mean rank proximity' score therefore showed a tendency to select incorrect contextual cues that were proximal rather than distal relative to the probed bird stimulus.

In the 'Ring Topology Test', we further test behavioural evidence for the spread of association between stimuli using a metric of 'Topological Distance', which we define as: the average link distance between the bird-bird and bird-scene choices made by each participant. For example, a correct pairing between bird '1' and '2', or between bird '1' and sofa 'A' would be given a rank score of 1, while an incorrect pairing between bird '1' and bird '3', or between bird '1' and sofa 'B' would be given a rank score of 2, and so on. Thus, if the mean 'Topological distance' score is low, participants are more likely to erroneously position proximal rather than distal stimuli as neighbours on the rink.

## MRS data collection and analysis

MRS spectra were acquired using a locally developed version of the CMRR Spectroscopy Package MEscher-GArwood Point RESolved Spectroscopy (MEGA-PRESS) sequence in two 2 x 2 x 2 cm$^3$ VOIs, with TE = 68 ms, TR = 1.5 s, flip angle 90 deg, ON editing pulse 1.9 ppm, OFF editing pulse 7.5 ppm. Water suppression was achieved using per-subject calibrated VAPOUR and dual-band MEGA editing. Outer volume suppression around each axis was interleaved with the water suppression module. Unsuppressed water signal was collected for internal water referencing. The first volume of interest (VOI) was centred bilaterally on the calcarine sulcus in visual cortex (V1). The second VOI was positioned in right LOC. For each measurement 320 spectra were acquired (160 edit-ON and 160 edit-OFF), resulting in an acquisition time of about 8 min for each VOI. Shimming was performed using the vendor's own algorithm, based on GRE B0 maps (GRE Brain)[76].

Glx and GABA+ concentrations were quantified using Gannet version 3.1.4[77], with water used as a reference. Gannet's standard pre-processing pipeline was used, which includes frequency and phase correction by spectral registration and line broadening. Individual ON and OFF subspectra were then averaged and edited spectra were generated by subtracting the averaged edit-OFF spectra from edit-ON spectra. Notably, the editing approach not only targets GABA but also other macromolecules at 3 ppm, therefore the concentrations of GABA + (GABA and macromolecules) are reported. Furthermore, the editing approach also targets nearby glutamate/glutamine resonances, providing an estimate of Glx concentration. However, it should be noted that the MEGA-PRESS uses parameters optimised for editing GABA and not glutamate. Therefore, compared to quantifying Glx using a PRESS sequence, test-retest reliability of the Glx is likely compromised when estimated with MEGA-PRESS[78,79]. Metabolite concentrations were quantified in pseudo-absolute molality units (approximating moles of GABA per kg of solute water) and relaxation and tissue-corrected[80]. Grey matter, white matter and CSF tissue fractions for determining tissue-corrected concentrations were obtained for both using SPM12.

Reliable model fits were achieved for 38 out of 41 V1 acquisitions (ATX: 20, PLC: 18) and for 37 out of 43 LOC acquisitions (ATX: 19, PLC: 18). In data where no reliable model fit was achieved, GABA+ was inestimable due to lipid contamination or low SNR. Data quality was quantified using the signal-to-noise ratio (SNR) and full-width-at-half-max (FWHM) of N-acetylaspartate (NAA) and fit error of the GABA+ peak provided by Gannet (Supplementary Fig. 5). The average GABA+ and Glx concentrations for the ATX and PLC groups were compared using two-tailed permutation tests. A null distribution of the expected difference was generated by permuting group labels (ATX, PLC) 10,000 times using MATLAB's random number generator.

## fMRI scan task

During the fMRI scan, the participant viewed the 11 bird stimuli encountered during learning, across four scan blocks. During 2 of these scan blocks, bird stimuli were interleaved with the 11 sofa contextual stimuli. Stimuli were presented via a computer monitor inside the scanner bore. On each trial, stimuli were presented for 1000 ms (Fig. 3a). The inter-trial interval was selected from a truncated gamma distribution with a mean of 2.3 s, minimum of 1.3 s and maximum of 14 s. To control for potential confounding effects of expectation suppression[81], all possible transitions between non-repeating stimuli were presented once in each block, in a fully randomised order. Participants performed a task incidental to the contrast of interest which involved identifying whether the presented stimuli were familiar or "oddball". Oddball stimuli were distorted or differently coloured versions of the stimuli encountered during training, and were randomly inserted into 9% of trials. Participants were instructed to press a button on an MR-compatible button box using their right index finger when they identified oddball stimuli, but not if the stimulus was familiar.

Participants received a monetary reward for each correct oddball identification. No feedback was given other than the total monetary reward after each block.

## Pupillometry acquisition, preprocessing, and analysis

During the fMRI scan task (see *fMRI scan task*) pupillometry data was collected. We measured pupil size using an EyeLink eyetracker (SR Research) sampled at 500 Hz. Pupillometry data was preprocessed using a standardised processing pipeline. Briefly, data was smoothed using a 120 ms Gaussian kernel, blinks were identified and removed by interpolating the pupillometry data across the blink period. Data were epoched around presentation of the visual stimuli. Trials on which the pupil measurement was lost for a period of >1000 ms were removed from the analysis. Participants were excluded from subsequent analysis if more than 50% of trials had been removed ($n = 10$ participants). Overall, pupillometry data from 34 out of 44 participants was included (ATX: 15, PLC: 19), with an average of $672.50 \pm 17.69$ 'regular' trials and $58.82 \pm 1.36$ 'oddball' trials (mean ± SEM).

To normalise pupil size for each scan block we expressed pupil size in terms of the percentage difference from the average pupil size across the block. To account for changes in pupil size during the fMRI scan task, each epoched pupil response was baseline-corrected. The baseline was defined as the mean of the normalised pupil size in the 250 ms window prior to stimulus presentation. To assess the effect of surprise on the pupil response, for each participant, we computed the difference between the epoched pupil response for 'oddball' vs 'regular' stimuli. To assess the difference in pupil response between the ATX and PLC groups, we used a cluster-based significance approach where we permuted subject labels x10,000 for each timepoint, to define the size threshold for a statistically significant cluster ($p < 0.05$)[82,83].

## fMRI imaging protocol

The fMRI scan task was performed inside a 3 T Magnetom Prisma (Siemens) using a 32-channel head and neck coil (Nova Medical Inc, Wilmington, MA) at the Oxford University Centre for Integrative Neuroimaging (University of Oxford). fMRI data was acquired using a multiband echo-planar imaging (EPI) sequence, with multiband factor 3, TR = 1.235 s, TE = 20 ms, flip angle = 65 degrees, field of view = 216 mm, and a voxel resolution of $2 \times 2 \times 2$ mm[3]. All scans were of axial orientation, angled to the long axis of the hippocampus, covering the whole brain. For each participant, a T1-weighted structural image was acquired to correct for geometric distortion and perform co-registration between EPIs, consisting of 192 1mm axial slices, in-plane resolution of 1×1 mm[2], TR = 1.9 s, TE = 3.97 ms, and field of view = 192 mm. A field map with dual echo-time images was also acquired (TE1 = 4.92 ms, TE2 = 7.38 ms, whole-brain coverage, voxel size 2x2x2 mm[3]). Physiological measures (cardiac pulse and thoracic movement) were collected during the scan but not used in analysis. Cardiac pulse was recorded using an MRI-compatible pulse oximeter. Thoracic movement was monitored using a custom-made pneumatic belt positioned around the abdomen.

## fMRI preprocessing and GLMs

Preprocessing of fMRI data was carried out in SPM12 (https://www.fil.ion.ucl.ac.uk/spm/). The dataset of one participant was excluded due to excessive motion. For the 43 participants included in the fMRI analysis, images were corrected for signal bias, realigned to the first volume of the block, corrected for distortion using fieldmaps, normalised to a standard EPI template and smoothed using a 5 mm full-width at half maximum Gaussian kernel. To remove low-frequency noise from the preprocessed data, a high-pass filter was applied to the data using SPM12's default settings. For each participant and for each scan block presenting bird stimuli, the resulting fMRI data was analysed in an event-related manner using three General Linear Models

(GLMs). Explanatory variables used a delta function to indicate the onset of a stimulus and were then convolved with the hemodynamic response function. In addition to the explanatory variables (EVs) of interest (described below), six additional scan-to-scan motion parameters produced during realignment were included in the GLM as nuisance regressors to account for motion-related artefacts in each task block.

The first GLM was used to identify brain regions that represent task stimuli (Supplementary Fig. 5b). 14 EVs were included per scan block. The first 'repetition suppression' EV was a parametric regressor corresponding to the number of trials between the current stimulus and the most recent previous presentation of the same stimulus. The regressor was subsequently z-scored. The next 11 EVs accounted for each of the different bird stimuli (bird 1–11). The remaining 2 EVs accounted for detected and undetected 'oddball' trials, respectively. All EVs were then convolved with the hemodynamic response function. The contrast of interest was the parametric regressor for repetition suppression, contrasted against baseline.

The second GLM was used to quantify the neural spread of association across the '11-node' underlying task structure. 14 EVs were included per scan block. The first EV was the parametric regressor corresponding to the link distance (nodes) between the presented stimulus and the preceding stimulus measured along the ring structure. The parametric regressor was according to an exponential function ($1-2^{[\text{link distance}]}$) to account for non-linearity in link distance encoding, and was subsequently z-scored. The next 11 EVs accounted for each of the different bird stimuli (bird 1-11). The remaining 2 EVs accounted for detected and undetected 'oddball' trials, respectively. All EVs were then convolved with the hemodynamic response function. The contrast of interest was the parametric regressor of link distance (Fig. 5b), contrasted against baseline.

The third GLM was used to quantify the neural spread of association using the output from the '6-node' neural network model. 17 EVs were included per scan block. Similar to the '11-node' GLM described above, the first EV was the parametric regressor corresponding to the link distance between the presented stimulus and the preceding stimulus. An additional parametric regressor was included, which corresponded to the link distance between the presented stimulus and the stimulus that was presented 2 trials back. The two parametric regressors were defined according to the peak firing rate of assemblies in the neural network model (Fig. 4c, middle panel), and subsequently z-scored. As the parametric regressors could only take into account stimuli 1 to 3 links away, 2 additional EVs were included to account for stimuli 4 or 5 links separated on the previous trial and the trial before the previous. As in the 11-node model, the remaining EVs account for the different bird stimuli (11 EVs) and detected and undetected oddball trials (2 EVs). All EVs were then convolved with the hemodynamic response function. The contrast of interest was the main of the two parametric regressors of link distance, contrasted against the baseline.

## fMRI analysis and statistics

The contrast of interest was entered into second-level random effects 'group' analyses to assess the main effect of link distance, and to assess the effect of 4 covariates: GABA+ in LOC, pupil response, and behaviour ('Overgeneralisation errors' and 'Mean rank proximity'). Second-level analyses were performed for each group separately (ATX, PLC), and for all participants combined. We set the cluster-defining threshold to $p < 0.01$ uncorrected before using whole-brain family-wise error (FWE) to correct for multiple comparisons, with the significance level defined as $p < 0.05$. To assess changes in BOLD in regions of interest where we predicted representation of task stimuli and a spread of association across memory (Fig. 5c), we performed small volume correction for multiple comparisons, with peak-level FWE correction at $p < 0.05$. For the small volume correction in hippocampus and

parahippocampus, we used bilateral anatomical masks segmented from several representative T1 images normalised to the MNI152 template (Supplementary Fig. 12a), followed by post hoc tests using unilateral hippocampal or parahippocampal masks. For small volume correction in LOC we used a 10 mm radius sphere centred on the group-average MRS voxel (Supplementary Fig. 5a).

To assess the relationship between fMRI cross-stimulus suppression and each of the 4 covariates of interest (GABA+ in LOC, pupil response, 'Overgeneralisation errors' and 'Mean rank proximity'), we ran the second-level random effects model with the inclusion of each covariate at the group level. For the GLM with GABA +, an additional covariate was included to account for differences in the concentration of Glx in LOC. Only participants with reliable MRS model fits were included, resulting in $n = 19$ participants in the ATX group and $n = 17$ participants in the PLC group. To assess the relationship between pupil response and fMRI, a pupil response covariate was defined as the average difference in pupil response to 'oddball' stimuli versus regular stimuli, across the time period where we observed a group difference between ATX and PLC (Fig. 3b). Covariates for the concentration of GABA+ and Glx in LOC were included to assess variance predicted by the pupil response, over and above variance explained by changes in GABA+ and Glx. Thus, data for a given participants was only included if pupillometry data quality met the inclusion criteria (defined above) and if MRS data resulted in reliable model fits, resulting in $n = 15$ participants in the ATX group and $n = 12$ participants in the PLC group. To assess the relationship between fMRI cross-stimulus suppression and behaviour, we included a covariate for the behavioural measures of overgeneralisation from the implicit memory test (Fig. 2f, g).

Furthermore, we used a region of interest (ROI) approach to visualise the relationships between fMRI cross-stimulus suppression and GABA+ in LOC, pupil response, and behaviour. To this end, we used anatomical ROIs for right/left parahippocampus and right/left hippocampus to extract beta parameter estimates for the parametric link distance regressor, and computed the Spearman rank correlation for each relationship (Fig. 6a, b, e, f; Supplementary Fig. 14).

**Recurrent spiking neural network model**

All simulations were carried out in C++ using Auryn (http://www.fzenke.net/auryn).

**Neuron Model.** The postsynaptic neurons were described by point leaky integrate-and-fire neurons with after-hyperpolarisation (AHP) current, whose membrane potential $V(t)$ evolved according to ref. 47:

$$\tau_m \frac{dV(t)}{dt} = (V_{rest} - V(t) + g_{AHP}(t)(V_{AHP} - V(t)) + g_{AMPA}(t)(V_{AMPA} - V(t)) + g_{GABA}(t)(V_{GABA} - V(t)) + g_{NMDA}(t)(V_{NMDA} - V(t)) \tag{2}$$

where $\tau_m = 30ms$ is the membrane time constant and $V_{rest} = -65mV$ is the resting membrane potential; $V_{AMPA} = V_{NMDA} = 0mV$ and $V_{GABA} = V_{AHP} = -80mV$ are the reversal potentials; $g_{AMPA}$, $g_{NMDA}$, and $g_{GABA}$ are the synaptic conductances for their respective channels, and $g_{AHP}$ is the conductance of the AHP channel. The neuron fires when its membrane potential crosses a fixed threshold of $V_{th} = -50mV$ below, after which it returns to its resting potential where it remains clamped for $\tau_{ref}^E = 5ms$ and $\tau_{ref}^I = 2.5ms$ of refractory period in excitatory and inhibitory neurons, respectively.

**Synapse Model.** The synapses were conductance-based. The AHP conductance follows the equation[47]:

$$\frac{dg_{AHP}(t)}{dt} = -\frac{g_{AHP}(t)}{\tau_{AHP}} + A_{AHP}S_{post}(t) \tag{3}$$

Where $\tau_{AHP} = 100ms$ is the characteristic time of the AHP channel, and $A_{AHP} = 5nS$ is the amplitude of increase in conductance due to a single postsynaptic spike. $S_{post}(t) = \sum_k \delta(t - t_{k,post}^*)$ is the postsynaptic spike train, where $t_{k,post}^*$ is the time of the $k^{th}$ spike of the postsynaptic neuron, and $\delta(.)$ is the Dirac's delta.

The synaptic conductance for $X$ (where $X$ can be $AMPA$, $GABA$, or $NMDA$) obeys the following equation[47]:

$$\frac{dg_X(t)}{dt} = -\frac{g_X(t)}{\tau_X} + \sum_{j \in X} w_j(t)S_j(t) \tag{4}$$

Here, the characteristic times are $\tau_{AMPA} = 5ms$, $\tau_{GABA} = 10ms$, $\tau_{NMDA} = 150ms$. $S_j(t) = \sum_k \delta\left(t - t_{k,j}^*\right)$ represents the presynaptic spike train, where $t_{k,j}^*$ is the time of the $k^{th}$ spike of neuron $j$.

*Plasticity Model:* The plasticity model employed in these simulations was the so-called "codependent" plasticity model[47] based on multiple empirical findings. Codependent synaptic plasticity rules depend on both spike times and synaptic currents. The codependent plasticity function $\phi$ of a synapse is represented as:

$$\phi(E, I, pre, post) \tag{5}$$

where $E$ and $I$ are the excitatory and inhibitory currents due to neighbourhood synaptic activity (including the synapse's own activity), and $pre$, $post$ are the spike timings of the pre- and postsynaptic neurons, respectively. $E$ represents the calcium influx through N-methyl D-aspartate (NMDA) channels to model excitatory currents and $I$ represents the chloride influx through gamma-aminobutyric acid (GABA) channels to model inhibitory currents. These currents are defined as follows:

$$\tau_E \frac{dE(t)}{dt} = -E(t) - \sum_k I_k^{NMDA}(t) \tag{6}$$

$$\tau_I \frac{dI(t)}{dt} = -I(t) + \sum_k I_k^{GABA}(t) \tag{7}$$

where $\tau_E = 10ms$ and $\tau_I = 100ms$ are the characteristic time constants for $E$ and $I$, respectively. $I_k^{NMDA}(t)$ and $I_k^{GABA}(t)$ are the currents flowing through the NMDA and GABA channels for the $k^{th}$ excitatory and inhibitory synapses, respectively.

The learning rule of the $j^{th}$ excitatory synapse onto the postsynaptic neuron is defined as:

$$\frac{dw_j(t)}{dt} = \phi_E\left(E_j(t), I(t); S_j(t), S_{post}(t)\right)$$
$$= \left[A_{LTP}x_j^+(t)S_{post}(t)E_j(t) - A_{LTD}y_{post}^-(t)S_j(t)w_j(t) - A_{het}y_{post}^E(t)S_{post}(t)(E_j(t))^2\right]\exp\left[-\left(\frac{I(t)}{I^*}\right)^\gamma\right] \tag{8}$$

where $A_{LTP} = 3 \times 10^{-4}$ and $A_{LTD} = 3 \times 10^{-5}$ are the learning rates of Hebbian long-term potentiation and long-term depression, respectively, and $A_{het} = 1.5 \times 10^{-8}$ is the learning rate of the heterosynaptic plasticity, which depends quadratically on the neighbouring excitatory currents. Importantly, in the co-dependent plasticity rules, the heterosynaptic plasticity term counterbalances the destabilising effects of excitatory potentiation. Without the heterosynaptic plasticity term the excitatory synaptic weights do not stabilise and reach a maximum possible value, with excitatory firing attaining implausible firing rates (Supplementary Fig. 9, Supplementary Fig. 10). Heterosynaptic plasticity therefore provides a stabilising mechanism to counteract the excitatory plasticity that enables: *(1)* embedding of sensory representations (both the PLC and ATX networks); *(2)* embedding of the associative memories (i.e. ring topology: ATX and

PLC); *(3)* the graded excitatory synaptic weights across assemblies (ATX model only).

$I^* = 200$ and $\gamma = 3$ represent the level and shape of the control of inhibitory activity over changes in excitatory weights (Supplementary Fig. 9, Supplementary Fig. 10). $S_j(t)$ and $S_{post}(t)$ are the pre- and post-synaptic spike trains, as described earlier. $x_j^+(t)$, representing the trace of the presynaptic spike train, follows:

$$\frac{dx_j^+(t)}{dt} = -\frac{x_j^+(t)}{\tau_+} + S_{post}(t) \qquad (9)$$

$y_{post}^-(t)$ and $y_{post}^E(t)$, representing the traces of the postsynaptic spike trains with different time scales, follow:

$$\frac{dy_{post}^-(t)}{dt} = -\frac{y_{post}^-(t)}{\tau_-} + S_j(t) \qquad (10)$$

and

$$\frac{dy_{post}^E(t)}{dt} = -\frac{y_{post}^E(t)}{\tau_E} + S_{post}(t) \qquad (11)$$

where $\tau_+ = 16.8 ms$, $\tau_- = 33.7 ms$, and $\tau_E = 100 ms$.

The inhibitory learning rule of the $j^{th}$ inhibitory synapse onto the postsynaptic neuron is defined as:

$$\frac{dw_j(t)}{dt} = \phi_I\left(E_j(t), I(t); S_j(t), S_{post}(t)\right)$$
$$= A_{ISP}E_j(t)[E_j(t) - \alpha I(t)][x_j(t)S_{post}(t) + y_{post}(t)S_j(t)] \qquad (12)$$

where $A_{ISP} = 1 \times 10^{-3}$ is the codependent inhibitory plasticity learning rate, $\alpha$ is the excitatory-inhibitory balance point, such that $\frac{E}{I} = \alpha$. $x_j(t)$ and $y_{post}(t)$ are the synaptic traces of the pre- and postsynaptic spike trains, respectively. They follow:

$$\frac{dx_j(t)}{dt} = -\frac{x_j(t)}{\tau_{STDP}} + S_{post}(t) \qquad (13)$$

and

$$\frac{dy_{post}(t)}{dt} = -\frac{y_{post}(t)}{\tau_{STDP}} + S_j(t) \qquad (14)$$

where $\tau_{STDP} = 20 ms$.

**Network and Simulation.** In the neural network model, we simulated a recurrent neural network of 1000 excitatory and 250 inhibitory neurons. E-E and I-E connections were plastic, while E-I and I-I connections were static. Connectivity was 5% between and across all neuron groups. Initial weights for the different synapse types were $w_{EE} = 0.25 nS$, $w_{EI} = 0.35 nS$, $w_{IE} = 0.31\ nS$, $w_{II} = 0.31\ nS$. Background network activity was maintained by connecting a group of 100 Poisson neurons firing at 1 Hz to the excitatory neuron group with a weight of $w_{input} = 0.1 nS$ and connection probability of 1.

**Cell Assembly Formation.** The cell assemblies *1–6* were formed by connecting six independent Poisson groups (representing external sensory stimuli) of 100 neurons each, firing at 2.5 Hz, to six groups of 100 excitatory neurons each. The synapses between the Poisson neurons and the excitatory neurons had a weight of $0.1 \pm 0.01 nS$. This resulted in an increase in network excitation, which was subsequently balanced by strengthening $w_{IE}$ (I-E connections) between groups of 25 inhibitory neurons to each of the formed excitatory cell assemblies to

$w_{IE} = 0.9 \pm 0.05 nS$. Hence, each assembly *1–6* was composed of 100 excitatory neurons and 25 inhibitory neurons.

**Inter-nodal Associations.** In both the placebo and atomoxetine neural network simulations, excitatory connections between formed nodes were strengthened by setting E-E connections between them to $w_{EE} = 0.43 \pm 0.02 nS$. However, in the atomoxetine simulations, these E-E connections were strengthened under decreased inhibitory firing, which was achieved by setting $w_{EI}$ to only 97% of its value, or $w_{EI} = 0.3395 nS$.

In the placebo simulations, this E-E strengthening was followed by setting inter-nodal I-E connections to $w_{IE} = 0.9 \pm 0.05 nS$. In the ato-moxetine case, these connections were only strengthened to $w_{IE} = 0.7 \pm 0.05 nS$, which was the minimum I-E strengthening required to prevent runaway excitation due to E-E potentiation triggered by the inter-nodal excitatory associations.

**Driving Activity of Assembly '1'.** To test the degree of coactivation, the activity in assembly *'1'* was driven externally for 3.3 s during the period marked "stimulus" (Fig. 4, centre plots) by increasing the weight of the relevant Poisson group representing the sensory input to $0.5 nS$.

### Reporting summary

Further information on research design is available in the Nature Portfolio Reporting Summary linked to this article.

## Data availability

All data generated and analysed during this study are included in the manuscript and supporting files. Source data are provided with this paper. Group-level data is available from the MRC BNDU Data Sharing Platform via https://doi.org/10.60964/BNDU-Z7QY-JP81. The following dataset was generated: fMRI data. MRS data. Pupillometry data. Behavioural data Source data are provided with this paper.

## Code availability

Upon publication the code used for data analysis will be available from the MRC BNDU Data Sharing Platform via https://doi.org/10.60964/BNDU-Z7QY-JP81. Upon publication the code used for the spiking neural network model will be available from the MRC BNDU Data Sharing Platform via https://doi.org/10.60964/BNDU-9B3H-A961, and is also available on https://github.com/p-rakriti/koolschijn_et_al.

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

## Acknowledgements

We would like to thank Chamith Halahakoon, Phil Cowen, Angharad De Cates, Beata Godlewska, Riccardo De Giorgi, Katherine Smith and Edoardo Ostinelli for enabling this study by providing medical cover. We would like to thank Douglas F. Tomé and Everton J. Agnes for their guidance and advice with earlier versions of the neural network model. We would like to thank Rob Froemke for helpful discussion when preparing the experiments. We thank Leonie Glitz and Valentina Mancini for comments on an earlier version of the manuscript. R.S.K. was supported by an EPSRC/MRC-funded studentship (EP/L016052/1). P.P. was supported by the Cambridge Trust, Trinity Henry Barlow Scholarship and Trinity Hall Brockhouse Scholarship. L.C. is supported by the Foundation for Science and Technology (FCT) (Portuguese State Budget: UID/PSI/01662/2020; Research fellowship: 2021.00415.CEECIND). W.T.C. is funded by the Wellcome Trust [225924/Z/22/Z]. H.C.B. is supported by a UKRI Future Leaders Fellowship (MR/W008939/1) and the Wellcome Institutional Strategic Support Fund. H.C.B. and J.X.O. are supported by the Medical Research Council (MR/W01971X/1). The study was supported by the NIHR Oxford Health Biomedical Research Centre (NIHR203316). The views expressed are those of the author(s) and not necessarily those of the NIHR or the Department of Health and Social Care. The Wellcome Centre for Integrative Neuroimaging is supported by core funding from the Wellcome Trust (203139/Z/16/Z and 203139/A/16/Z). This research was funded in part by the Wellcome Trust. For the purpose of open access, the author(s) have applied a CC BY public copyright license to any Author Accepted Manuscript version arising from this submission.

## Author contributions

All authors contributed to the preparation of the manuscript. R.S.K., J.X.O., and H.C.B. designed the study. R.S.K. and H.C.B. acquired ethics for the study. L.C. and R.S.K. prepared and administered the double-blind procedure. M.B. provided clinical support for the study. R.S.K. acquired the data. W.C. and R.S.K. developed the MRS sequence and analysed the MRS data. R.S.K. analysed the behavioural data, pupillometry data and fMRI data with supervision from J.X.O. and H.C.B. X.P. assisted with the fMRI analyses. P.P. generated all neural network simulations with supervision from T.P.V. R.S.K., P.P., and H.C.B. prepared the figures.

## Competing interests

M.B. has received travel expenses from Lundbeck for attending conferences and has acted as a consultant for J&J, Novartis, Boehringher and CHDR. He previously owned shares in P1vital Ltd. All other authors declare no competing interests.
