## [Transparent Peer Review file · Nature Communications]

Noradrenaline causes a spread of association in the hippocampal cognitive map

Corresponding Author: Dr Renée Koolschijn

Version 0:

Reviewer comments:

Reviewer #1

(Remarks to the Author)

The manuscript titled 'Noradrenaline causes a spread of association in the hippocampal cognitive map' presents a variety of new empirical data that the authors argue, together, indicate that elevated noradrenergic (NE) tone promotes generalization of statistical learning. Human subjects were either administered ATX, an NE-reuptake inhibitor, or a placebo before viewing a sequence of images of two birds and scenes. The combinations of birds and scenes presented across stimuli were constructed according to a pre-defined grammar. Importantly, in this grammar, each bird had only two others it could appear with but, therefore, also had second degree neighbors—birds that were never shown with a particular bird but were shown with the birds that had—enabling implicit and explicit assessments of the associative strength of 2nd degree associations (e.g., via transitive inference). The key observation supporting the main conclusion is shown in Figure 2F, indicating that overgeneralization errors are significantly more likely for those who received ATX than for those who received the placebo. The remainder of the results are used to support the efficacy of the drug in modulating NE levels, modeling of the main result, and evidence of brain activity that aligns with a prediction of the model.

The described work is elegant and creates a multi-faceted portrait of how modulating NE tone influences neural processing and resulting learning. Overall, I'm impressed at the experimental design and see the pattern of results as being fairly cohesive and moderately convincing. The final story that the authors tell from the data aligns well with the evidence presented. If the result indeed proved reliable, the finding would be impactful and timely. It is of high significance to understand the how modulatory signaling influences the nature of learned representations and the cognitive map is a framework that is having a moment of increased attention. This work addresses both the influence of neuromodulation and a mechanism of cognitive map control. Though I don't see overt flaws with the work, my enthusiasm remains tempered because of two main issues: 1) minimal data to support primary claims, and 2) little effort to rule out competing hypotheses. These are unpacked below.

Major issues –

Minimal data to support primary claims. The entire story that NE increases generalization in the cognitive map is supported by a single empirical comparison (Fig 2F). This comparison appears to have been done rigorously but it is not supported with any secondary analyses of any sort. Given that no other differences were observed between the ATX and placebo conditions, I wonder if how robust this one comparison is. Looking at the raw data points that are plotted in Fig 2F, it seems that only a subset (60%?) of the subjects in the ATX condition exhibited the effect at all. I don't want to prescribe one versus another way to reaffirm that this is a robust effect, but I'll say that this one comparison leaves room for concern that the effect may not be robust. Relatedly, for the result about repetition suppression as a function of distance in the ring of stimuli, a significant pattern is observed in one hippocampus for ATX but this was not significantly different from the placebo condition. Though this is something, it isn't fully convincing. Relatedly, this lack of difference should not be buried in the extended materials.

Little effort to rule out competing hypotheses. No competing hypotheses were named in the manuscript. Though I'll agree that the main narrative offered for interpreting the data makes sense, it does not seem to me that it is unique in its ability to account for the data. For example, the modeling presented is one approach to generate spreading associations. Other approaches could also likely work. Likewise, related to the point above, there is little consideration of other possible explanations for the difference in implicit error rate.

Minor issues –

It isn't clear how 'Overgeneralization errors (%)' were calculated. The methods indicates that it is the percentage of trials that proximal context is chosen minus the percentage of the trials where any other error was made. Either my understanding of this is wrong or participants are selecting the proximal context even more often than they are selecting the correct context. To unpack – six contexts are shown (16.67% chance), one is correct and this is chosen ~30% of the time (Fig 2), I imagine one of the remaining five is a proximal context (by chance 1 in 5, 20%, of errors should be to a proximal context) while the other four are non-proximal (80% of errors should be this by chance). Fig 2F shows that, even in the for the placebo condition, 'overgeneralization errors' are about 0%, which I understand means the proximal context is chosen as often as 'any other error was made. This means that 50% of the errors, even in the placebo condition are to the proximal context. Given that ~70% of trials were errors, 50% of those would be 35% of all trials – a value higher than the percent of correct trials. I imagine I'm misunderstanding something – hopefully the authors can spot what and clarify as necessary to avoid such distracting misunderstanding.

The timing of the elevated pupil size is extremely slow relative to the task design. The significant difference is only observed a full trial after an oddball.

Fig 1F(top) DABEST representation of data in, using normal distributions, is not well suited to data the data which is not normally distributed.

The number of participants and/or degrees of freedom should be listed near where stats are reported.

Reviewer #2

(Remarks to the Author)

In this study, Koolschijn et al. investigated how noradrenaline (NA) affects forming implicit associations in a learning task in human volunteers. They used the NA reuptake inhibitor atomoxetine in a double-blind placebo-controlled design. Participants performed a smart associative learning task featuring both an explicit and an implicit learning component. While the explicit learning was unaffected by the treatment, atomoxetine increased over-generalization errors in the implicit domain. Physiological markers of increased NA (GABA tone, pupil size) predicted fMRI signs of this over-generalization effect, which in turn predicted behavioral effects. A simple network model could account for the observed 'spread of associations'. This is an unusually clear model and presentation of a complex and behaviorally relevant concept, with high explanatory power. I only have a few questions and comments.

Major points

1. The correlations between physiological signs of drug effect and fMRI signs of 'spread', as well as between the fMRI and behavioral signatures suggest that there might be a correlation directly between pupil/GABA tone and over-generalization behavior. Was this tested? If this was feasible to test, it would be interesting to report it.
2. It would be worthwhile to elaborate on the rationale behind focusing on LOC (is that 'lateral occipital complex'?). Was a change in GABA tone not expected, as well as observed elsewhere?
3. The Authors tend to report direct statistical comparisons of the groups, which is the better approach than saying 'significant in A but not significant in B'. I did not find this for the correlations of the physiological markers and 'neural spread of associations' (lines 405-407, 417-419). Was there a significant difference in the direct comparisons? In this regard, the sample sizes are also of relevance; since there were some exclusions according to the Methods, it would make sense to provide all actual sample sizes for the statistical comparisons in the figure legends.
4. I found the correlation results pertaining to Figure 5 very exciting. At the same time, I was somewhat confused by the figure. What exactly is plotted in the scatter plots in Fig.5E, G and H? Is every dot representing a single participant? If so, why are the plots 'illustrative' – is this a selection of the data? If so, it would be better to present all data in the figure.
5. The Methods state that participants were trained to criterion. Did learning time differ in the two groups? Did learning time show any correlation with the results (behavioral, neural, physiological signs)?

Minor points

1. Figure legend for panel 1F: subpanel assignments (e.g. upper right) do not seem to fit the image.
2. 'Runaway excitation' suggests an uncontrolled processes: I suggest rephrasing this.

Disclaimer: Not having performed those methods myself, I am not in a good position to judge the methodology for the MRS, and I could only assess the broad approaches with respect to the fMRI methodologies.

I believe that openness and transparency increase the fairness of peer review. Therefore, I am signing my reviews.
Balazs Hangya

Reviewer #3

(Remarks to the Author)

In this paper, the authors studied the effect of noradrenaline on the association of memory items in the formation of the hippocampal cognitive map of human patients. To this end, the authors designed a behavioral task in which the patients learn the association between bird-bird pairs and contextual cues. Presentations of these images had a virtual ring structure, "- bird - cue - bird - cue -..." The authors showed that an increasing level of noradrenaline causes the overgeneralization of the bird-bird association over a long distance along the ring. The authors constructed a computational network model to suggest that the overgeneralization is caused by the reduction of inhibitory plasticity by the noradrenergic level increase.

The experimental paradigm seems to be novel, and the topic is interesting. However, I am unsure whether the results shown here are qualitatively and quantitatively sufficient to warrant publication in Nat Comm. The quantitative information about the overgeneralization is also limited.

Major concerns:

Several studies have shown that memory engrams encoding sensory and behavioral experiences that occur in a temporal vicinity tend to overlap (e.g., Mitashita et al., Nature 1988; Cai et al., Nature 2016). In addition, several models have been proposed to account for the overlapping cell assemblies (e.g., Griniasty et al., Neural Comput 1993; Haga and Fukai, Phys Rev Lett, 2019; Burns et al., eNeuro 2022). The authors, however, simply assumed that cell assemblies encoding memory items are orthogonal and mutually non-overlapping. This setting of the model looks over-simplified. What experimental data justify this assumption?

Related to the above comment, there are no quantitative comparisons between the model and activity data (e.g., in Fig. 4), which also limits the validity of the proposed model and inhibitory mechanism of overgeneralization.

Is heterosynaptic plasticity in Eq. 7 crucial for the bird-bird association? How is overgeneralization affected by the heterosynaptic plasticity?

The explanation of network simulations is confusing. How E-E and I-E connections were modified or fixed in the simulations is hard to understand. For instance, in the Inter-nodal Association, the authors mention w_{EE} was fixed at the same values, but w_{EI} was set at different values in the placebo and atomoxetine conditions. Does this imply that the excitatory and inhibitory plasticity rules were not applied to these synapses? Then, which synapses were modifiable?

Version 1:

Reviewer comments:

Reviewer #1

(Remarks to the Author)

The revisions and responses to all my comments were thoughtful and thorough. I have no further concerns of substance. I wish the authors and their manuscript continued success.

(Remarks on code availability)

Reviewer #2

(Remarks to the Author)

The Authors were responsive to my points. I think this is an exciting study and I have no further concerns.

I believe that openness and transparency increase the fairness of peer review. Therefore, I am signing my reviews.
Balazs Hangya

(Remarks on code availability)

Reviewer #3

(Remarks to the Author)

I appreciate that the authors have revised their manuscript extensively in response to my comments. The novel simulation results and modifications made in the revised manuscript clarified all my concerns. I have no further comments on the manuscript. I feel this manuscript presents an interesting piece of work.

(Remarks on code availability)

I had no time to test the code.

Point-by-point response to Reviewers' comments

We would like to start by thanking the three reviewers for their constructive comments. We have addressed all of your comments in turn below. As you can see, in the revised manuscript we include new analyses and simulations of our computational model, together with substantial changes to the text. We hope you agree that these changes have significantly improved the manuscript. Below we have copied all reviewers' comments in blue, interleaved with our responses in black.

Reviewer #1 (Remarks to the Author):

The manuscript titled 'Noradrenaline causes a spread of association in the hippocampal cognitive map' presents a variety of new empirical data that the authors argue, together, indicate that elevated noradrenergic (NE) tone promotes generalization of statistical learning. Human subjects were either administered ATX, an NE-reuptake inhibitor, or a placebo before viewing a sequence of images of two birds and scenes. The combinations of birds and scenes presented across stimuli were constructed according to a pre-defined grammar. Importantly, in this grammar, each bird had only two others it could appear with but, therefore, also had second degree neighbors—birds that were never shown with a particular bird but were shown with the birds that had—enabling implicit and explicit assessments of the associative strength of 2nd degree associations (e.g., via transitive inference). The key observation supporting the main conclusion is shown in Figure 2F, indicating that overgeneralization errors are significantly more likely for those who received ATX than for those who received the placebo. The remainder of the results are used to support the efficacy of the drug in modulating NE levels, modeling of the main result, and evidence of brain activity that aligns with a prediction of the model.

The described work is elegant and creates a multi-faceted portrait of how modulating NE tone influences neural processing and resulting learning. Overall, I'm impressed at the experimental design and see the pattern of results as being fairly cohesive and moderately convincing. The final story that the authors tell from the data aligns well with the evidence presented. If the result indeed proved reliable, the finding would be impactful and timely. It is of high significance to understand the how modulatory signaling influences the nature of learned representations and the cognitive map is a framework that is having a moment of increased attention. This work addresses both the influence of neuromodulation and a mechanism of cognitive map control. Though I don't see overt flaws with the work, my enthusiasm remains tempered because of two main issues: 1) minimal data to support primary claims, and 2) little effort to rule out competing hypotheses. These are unpacked below.

We would like to thank the reviewer for their complimentary remarks. We are grateful to the reviewer for highlighting how the study creates a multi-faceted portrait for how noradrenergic tone influences neural processing and learning. We agree that there is a pressing need to understand neuromodulatory signalling at a mechanistic and computational level, and believe our findings contribute to this endeavour. We would like to also thank the reviewer for constructive suggestions for improving the manuscript. Below we outline how we have implemented all recommendations.

Major issues –

1. Minimal data to support primary claims. The entire story that NE increases generalization in the cognitive map is supported by a single empirical comparison (Fig

2F). This comparison appears to have been done rigorously but it is not supported with any secondary analyses of any sort. Given that no other differences were observed between the ATX and placebo conditions, I wonder how robust this one comparison is. Looking at the raw data points that are plotted in Fig 2F, it seems that only a subset (60%?) of the subjects in the ATX condition exhibited the effect at all. I don't want to prescribe one versus another way to reaffirm that this is a robust effect, but I'll say that this one comparison leaves room for concern that the effect may not be robust.

Thank you for this important point. In the revised manuscript we now provide secondary analyses to demonstrate that the effect shown in Figure 2F is robust to alternative mathematical definitions of “overgeneralisation error”.

As a reminder, in the original manuscript (Figure 2F) we quantified overgeneralisation errors on the ‘*Implicit memory*’ test (Figure 2B) as “*the percentage of trials where participants erroneously link a bird stimulus to a proximal (neighbouring) contextual cue in the underlying ring structure*”. For example, this equates to estimating the percentage of trials where sofa B was selected when prompted with bird 1, minus the percentage of trials where any other error was made. Under this original mathematical definition, we demonstrated significant evidence for overgeneralisation errors in the ATX group ($p=0.014$, bootstrap-test) and significantly more overgeneralisation errors in ATX compared to PLC (ATX - PLC: $p=0.048$, permutation test) (Figure 2F).

To address your concerns, in the revised manuscript we include two new measures of overgeneralisation. These two new measures are described below. In each case, we show a significant increase in overgeneralisation in the ATX compared to the PLC group, suggesting the original effect reported in Figure 2F is robust.

The first new measure of overgeneralisation is mathematically quantified as the ‘*Mean rank proximity*’ of the chosen context relative to the correct context in the underlying ring structure, across all error trials in the ‘*Implicit memory*’ test (Figure 2B). Thus, if the *Mean rank proximity* score is low, participants are more likely to erroneously pick contextual stimuli that are close rather than distal from the target context, suggesting evidence for overgeneralisation. As shown in the new Figure 2G below, the *Mean rank proximity* score is significantly lower in the ATX group than would be expected by chance ($p<0.001$, bootstrap test) and significantly lower in the ATX group compared to the PLC group (PLC – ATX: $p=0.031$, permutation test). Importantly, for this alternative quantification of behavioural overgeneralisation, we again observe a significant correlation with our neural measure for overgeneralisation in hippocampus in the ATX group (SVC with parahippocampus-hippocampus ROI: $t_{20}=6.16$, $p=0.002$, MNI coordinates; *Post hoc* tests with SVC in right hippocampus ROI $t_{20}=5.92$, $p=0.001$), and a significant between group difference (ATX - PLC, SVC with parahippocampus-hippocampus ROI, $t_{39}=4.51$, $p=0.011$, MNI coordinates; *Post hoc* tests SVC with right hippocampus, $t_{39}=4.03$, $p=0.013$). These results are now presented in the revised Fig. 5.

Notably, compared to the original behavioural definition of overgeneralisation, the *Mean rank proximity* score is more sensitive to the underlying topology of the ring structure and can be considered more comparable with the parametric modulator used for the fMRI analyses shown in Figure 5. Moreover, for the ATX group, the distribution of *Mean rank proximity* scores is not bimodal, suggesting that when using this more sensitive measure of overgeneralisation, the

effect of overgeneralisation is observed across the ATX group and not merely confined to a subset of participants.

The second new measure of overgeneralisation considers a '*Topological distance*' metric which derives from a different behavioural test. During the experiment participants were not made explicitly aware of the underlying task structure, including the ring topology. At the very end of the experiment, after all memory tests were completed on Day 5, participants were given the opportunity to construct the ring topology by placing each bird pairing in the correct scene ('*Ring topology task*'). At the time of the original manuscript submission we simply had not analysed data deriving from this final '*Ring topology task*'. In the revised manuscript we include analysis of the data from this final task in the new Supplementary Figure 4 where we estimate a *Topological distance* score. We define *Topological distance* as the average link distance between the bird-bird and bird-scene choices made by each participant. For example, a correct pairing between bird '1' and '2', or between bird '1' and sofa 'A' would be given a rank score of 1, while an incorrect pairing between bird '1' and bird '3', or between bird '1' and sofa 'B' would be given a rank score of 2, and so on. Thus, if the mean topological distance score is low, participants are more likely to erroneously position proximal rather than distal stimuli as neighbours on the ring, providing evidence for overgeneralisation that is constrained by the underlying topology of the task.

Consistent with performance on the '*Explicit memory*' and '*Implicit memory*' tests (Figure 2C-D), on the '*Ring topology task*' we observe no significant difference in knowledge for the learned associations (ATX – PLC: $p=0.487$, permutation test) or for the implicit contextual associations (ATX – PLC, $p=0.259$, permutation test) (new Supplementary Figure 4B-C). However, when accounting for errors in estimating the mean *Topological distance* across the entire reconstructed topology, we observed a significant group difference where participants that performed learning under elevated noradrenaline (the ATX group) were more likely to arrange stimuli according to the underlying task topology (ATX – PLC: $p=0.027$, permutation test) (new Supplementary Figure 4D). Together with Figure 2F and 2G, this new behavioural measure again demonstrates that a single dose of atomoxetine increases overgeneralisation errors in behaviour, where overgeneralisation is constrained by the underlying task structure or cognitive map.

For the reviewers convenience, we have copied the revised and new figures below, with edited and new text shown in **bold**. We hope the reviewer agrees that these additional analyses demonstrate that the key behavioural effect is robust.

Figure 2 | Elevated noradrenaline during learning increases overgeneralisation in behaviour

A Schematic of example trial in the ‘Explicit memory test’ performed on day 5, where participants were tested on the bird-bird associations learned on day 1. **B** Schematic of example trial in the ‘Implicit memory test’ performed on day 5, where participants were required to recall which contextual cue (sofa) was incidentally paired with each bird stimulus. **C-D,F,G** Upper: Bootstrap-coupled estimation (DABEST) plots. Black dot, mean; black ticks, 95% confidence interval; filled curve, sampling error distribution. Lower left: memory accuracy (mean +/- SEM). Lower right: null distribution of the group differences generated by permuting subject labels, purple dot: true group difference. **C** Average memory accuracy in the ‘Explicit memory test’ was relatively high and there was no significant difference in overall accuracy between groups (ATX – PLC ($n=22:22$): permutation test: $p=0.421$). **D** Average memory accuracy in the ‘Implicit memory test’ was relatively low: on average participants made errors on 75.4% of trials. There was no significant difference in overall accuracy between groups (mean: ATX: 75.3%; PLC: 75.6%; ATX – PLC ($n=22:22$): permutation test: $p=0.498$). **E** Schematic illustrating the predicted distortion in the underlying cognitive map, where a ‘spread of association’ leads to overgeneralisation in participants’ implicit memory. **F** ‘Overgeneralisation errors’ in the ‘Implicit memory test’ were defined as the percentage of error trials where, given the probe, the chosen contextual stimulus was neighbouring to the correct contextual stimulus. The ATX group made significantly more ‘Overgeneralisation errors’ than expected by chance ($p=0.014$, effect size computed from 10,000 bias-corrected bootstrapped resamples²⁷), and significantly more than the PLC group (ATX – PLC ($n=22:22$): permutation test $p=0.048$). **G** ‘Mean rank proximity’ in the ‘Implicit memory test’ was defined as the average distance in links between the probe and erroneously chosen contextual stimuli. For instance, a choice for sofa ‘B’ with probe ‘1’ would be ranked as ‘2’ whilst a choice for sofa ‘D’ would be ranked as ‘4’. ‘Mean rank proximity’ across all error trials in the implicit memory test in the ATX group was significantly lower than expected by chance (ATX: $n=21$, $p<0.001$, effect size computed from 10,000 bias-corrected bootstrapped resamples²⁷) and significantly lower than in the PLC group (ATX – PLC ($n=22:22$): permutation test: $p=0.031$).

Supplementary Figure 4 | Ring topology task: elevated noradrenaline during learning does not affect explicit and implicit memory accuracy but does affect estimates of topological distance.

A Schematic of the Day 5 “*Ring topology task*”, in which participants were asked to piece together all explicit and implicit associations experienced on Day 1, by dragging two bird stimuli to each scene to construct the entire underlying topology. **B-D** Upper: Bootstrap-coupled estimation (DABEST) plots. Black dot, mean; black ticks, 95% confidence interval; filled curve, sampling error distribution. Lower left: memory accuracy (mean +/- SEM). Lower right: grey: null distribution of the group differences generated by permuting subject labels; purple dot: true group difference. **B** Overall accuracy for the explicitly learned bird-bird associations in the *Ring topology task*. No significant difference in explicit memory accuracy was observed between groups (ATX – PLC (n=21:22): permutation test p=0.487). **C** Overall accuracy for the implicit bird-scene associations in the *Ring topology task*. No significant difference in implicit memory accuracy was observed between groups (ATX – PLC (n=22:21): permutation test p=0.259). **D** Overall ‘*Topological distance*’ on the *Ring topology task*, defined as the average link distance between the bird-bird and bird-scene choices made by each participant. For example, a correct pairing between bird ‘1’ and ‘2’, or between bird ‘1’ and sofa ‘A’ would be given a rank score of 1, while an incorrect pairing between bird ‘1’ and bird ‘3’, or between bird ‘1’ and sofa ‘B’ would be given a rank score of 2, and so on. A significant group difference was observed for the *Topological distance* score, with a lower score for the ATX group suggesting evidence for overgeneralisation constrained by the underlying topology of the task (excluding participants who correctly remembered the entire cognitive map, ATX – PLC (n=21:16): permutation test p=0.027).

2. Relatedly, for the result about repetition suppression as a function of distance in the ring of stimuli, a significant pattern is observed in one hippocampus for ATX but this was not significantly different from the placebo condition. Though this is something, it isn’t fully convincing. Relatedly, this lack of difference should not be buried in the extended materials.

We would like to thank the reviewer for raising this important point. We note that in the original analysis, no significant difference was observed between ATX and PLC for the main effect of neural spread of association which we now show in the main manuscript (Figure 5E), with a

non-significant trend in parahippocampus (Figure 5F). We also take this opportunity to emphasise to the reviewer that we do observe a significant group difference in hippocampus between ATX and PLC for the correlation between repetition suppression and behaviour (now Figure 6A; $t_{20}=6.16$, $p=0.002$). Moreover, this significant group difference in hippocampus between ATX and PLC for the correlation between repetition suppression and behaviour remains significant for alternative behavioural measures of overgeneralisation (new Figure 6B). We note that across the whole brain, the significant group difference is highly selective and only observed in the hippocampus. This result demonstrates that only in drug group (ATX) are the behavioural measures of overgeneralisation predicted by the neural measures of overgeneralisation, suggesting a meaningful relationship between brain and behaviour in response to drug.

Furthermore, following the suggestion from *Reviewer #3* below, in the revised manuscript we also provide direct quantitative comparison between the computational model and the data. Specifically, we assess fMRI cross-stimulus suppression as a function of distance in the underlying ring using a parametric regressor constructed from the peak firing rate of assemblies in the computational model, after the associations between cell assemblies have been embedded and plasticity rules given time to apply under atomoxetine (Figure 4C, ATX model). We then use this model-derived parametric regressor to predict fMRI cross-stimulus suppression. In the ATX group, the model derived parametric regressor revealed significant overgeneralisation in the hippocampus (new Figure 5G), comparable to the original analysis (now Figure 5D). In addition, the model derived parametric regressor revealed a significant group difference in hippocampus, with greater overgeneralisation in the ATX compared to the PLC group (new Figure 5I). Arguably, this model-derived parametric regressor more appropriately captures the graded co-activity around the underlying cognitive map.

We have edited the manuscript to account for the above and present the new data analysis in the new Figure 5 which we copy below for the reviewers' convenience. We hope the reviewer agrees that these revisions provide clarity and further evidence to strengthen the overall claim.

Figure 5 | Elevated noradrenaline induces a neural spread of association in the hippocampus and parahippocampus

A Schematic illustrating the hypothesised regions involved in representing the memory map (Fig. 1A). **B** Schematic of the cross-stimulus suppression (XSS) fMRI contrast. **C-E** Evidence for neural spread of association in ATX, using an exponential function as a parametric regressor to model link distance between stimulus pairs, across all possible link distances in a 11-node ring (Fig. 1A). **C** T-statistic map showing XSS for spread of association in the ATX group, with significant effect in the right hippocampus and right parahippocampal cortex (SVC with parahippocampus-hippocampus ROI: $n=22$, $t_{21}=4.56$, $p=0.032$, MNI coordinates, Supplementary Table 3). *Post hoc* tests revealed significant effects in right parahippocampus ($n=22$, $t_{21}=4.56$, $p=0.005$), left parahippocampus ($n=22$, $t_{21}=3.51$, $p=0.017$), and right hippocampus ($n=22$, $t_{21}=4.35$, $p=0.017$). **D** Significant XSS spread of association in the ATX group. Lower: individual β parameter estimates obtained from the right hippocampus (R-HC: $n=22$, $p=0.003$, effect size computed from 10,000 bias-corrected bootstrapped resamples²⁷) and right parahippocampus (R-PHC: $n=22$, $p=0.0012$, effect size computed from 10,000 bias-corrected bootstrapped resamples²⁷) (mean \pm SEM). Upper: DABEST plots. Black dot, mean; black ticks, 95% confidence interval; filled curve, sampling error distribution. **E** T-statistic map showing no significant group differences for the spread of association effect (SVC with parahippocampus-hippocampus ROI, ATX - PLC ($n=22:21$): $t_{41}=2.35$, $p=0.595$, MNI coordinates). **F** *Post hoc* tests revealed a trend towards a significant group difference (ATX-PLC) in right parahippocampus for the spread of association effect (ATX - PLC ($n=22:21$): $p=0.041$, permutation test). Lower: individual β parameter estimates obtained from the right parahippocampus (mean \pm SEM). Upper: DABEST plots. Black dot, mean; black ticks, 95% confidence interval; filled curve, sampling error distribution.

G Similar to D, but for a 6-node model. **H** Similar to E, but for a 6-node model. **I** Similar to F, but for a 6-node model.

distribution. G-I Quantitative comparison between the ATX neural network model (Figure 4C) and data. XSS for spread of association in the ATX group was modelled using the peak firing rate of assemblies in the model (Figure 4C, middle) as a predictor for co-activation across link distance 1-3. G In the ATX group a significant spread of association was observed in right hippocampus (ATX: $n=22$, $p=0.024$, effect size computed from 10,000 bias-corrected bootstrapped resamples²⁷), with a trend in right parahippocampus (ATX: $n=22$, $p=0.055$, bootstrap test). H T-statistic map showing a trend towards a group difference (ATX–PLC) for the spread of association effect (SVC with parahippocampus-hippocampus ROI: ATX – PLC ($n=22:21$), $t_{41}=2.96$, $p=0.136$, MNI coordinates, Supplementary Table 5). I *Post hoc* tests revealed a significant group difference (ATX–PLC) in right hippocampus for the spread of association effect (ATX – PLC ($n=22:21$): $p=0.041$, permutation test). Lower: individual β parameter estimates obtained from the right hippocampus (mean \pm SEM). Upper: DABEST plots. Black dot, mean; black ticks, 95% confidence interval; filled curve, sampling error distribution.

Figure 6 | Neural spread of association predicts behavioural measure of overgeneralisation and is predicted by physiological measures of reduced inhibitory tone

A Left: In ATX, the neural measure of spread of association in hippocampus and parahippocampal cortex predicted ‘Overgeneralisation errors’ reported in behaviour (SVC with parahippocampus-hippocampus ROI, $n=22$, $t_{20}=6.16$, $p=0.002$, MNI coordinates, Supplementary Table 6). *Post hoc* tests revealed a significant effect in right hippocampus (SVC with right hippocampus ROI, $n=22$, $t_{20}=6.16$, $p=0.001$). Right: correlation plot using extracted parameter estimates from right hippocampus (Supplementary Fig. 11A) (Spearman correlation: $n=22$, $r_{20}=0.682$, $p>0.001$). **B** Left: In ATX, the neural measure of spread of association in hippocampus and parahippocampal cortex predicted ‘Mean rank proximity’ reported in behaviour (SVC with parahippocampus-hippocampus ROI, $n=22$, $t_{20}=6.16$, $p=0.002$, MNI coordinates, Supplementary Table 7). *Post hoc* tests revealed a significant effect in right hippocampus (SVC with right hippocampus ROI, $n=22$, $t_{20}=5.92$, $p=0.001$). Right: correlation plot using extracted parameter estimates from right hippocampus (Spearman correlation: $n=22$, $r_{20}=-0.595$, $p=0.004$). **C** Group difference (ATX – PLC) for the correlation between the neural spread of association (XSS) and behavioural measure of ‘Overgeneralisation errors’ (ATX – PLC ($n=22:21$): SVC with parahippocampus-hippocampus ROI, $t_{41}=4.51$, $p=0.011$, MNI

coordinates, Supplementary Table 8). *Post hoc* tests revealed a significant effect in right hippocampus (ATX – PLC (n=22:21): SVC with right hippocampus ROI, $t_{41}=4.51$, $p=0.004$). D Group difference (ATX – PLC) for the correlation between the neural spread of association (XSS) and behavioural measure of ‘Mean proximity rank’ (ATX – PLC, SVC with parahippocampus-hippocampus ROI, $t_{41}=4.51$, $p=0.011$, MNI coordinates, Supplementary Table 9). *Post hoc* tests revealed a significant effect in right hippocampus (ATX – PLC (n=22:21): SVC with right hippocampus, $t_{41}=4.03$, $p=0.013$). E Left: In ATX, the concentration of GABA⁺ in LOC predicted the neural measure of spread of association in hippocampus and parahippocampal cortex (SVC with parahippocampus-hippocampus ROI, n=19, $t_{16}=4.70$, $p=0.047$, MNI coordinates, Supplementary Table 10). *Post hoc* tests revealed a significant correlation in right and left parahippocampus (SVC with left parahippocampus ROI, n=19, $t_{16}=4.70$, $p=0.008$; SVC with right parahippocampus ROI, n=19, $t_{16}=4.07$, $p=0.023$). Right: Correlation plot using extracted parameter estimates in right parahippocampus to illustrate effect reported in whole-brain map (Spearman correlation: n=19, $r_{17}=-0.661$, $p=0.003$). F Left: In ATX, positive trend between the pupil dilation response to surprising stimuli (Fig. 3B) and the neural spread of association (SVC with parahippocampus-hippocampus ROI, n=15, $t_{11}=5.15$, $p=0.066$, MNI coordinates, Supplementary Table 11). *Post hoc* tests revealed a significant correlation in right hippocampus (SVC with right hippocampus, n=15, $t_{11}=5.15$, $p=0.023$). Right: correlation plot using extracted parameter estimates from right hippocampus (Spearman correlation: n=15, $r_{13} = 0.511$, $p=0.054$). T-statistic maps are thresholded at $p < 0.01$ uncorrected, for visualisation purposes only.

3. Little effort to rule out competing hypotheses. No competing hypotheses were named in the manuscript. Though I’ll agree that the main narrative offered for interpreting the data makes sense, it does not seem to me that it is unique in its ability to account for the data. For example, the modeling presents is one approach to generate spreading associations. Other approaches could also likely work. Likewise, related to the point above, there is little consideration of other possible explanations for the difference in implicit error rate.

We thank the reviewer for raising this point. We agree that other mechanisms could generate the overgeneralisation effect and we are grateful for the opportunity to further explore this possibility.

In our original data analysis we considered two hypotheses:

Data hypothesis #1: Learning under elevated noradrenaline quantitatively strengthens memory encoding/consolidation of each component association, resulting in enhanced representation of the entire ring topology which promotes overgeneralisation behaviour.

Data hypothesis #2: Learning under elevated noradrenaline does not quantitatively strengthen memory encoding/consolidation of each component association, but does increase the graded co-activity in the underlying representation of the ring topology (or cognitive map) to promote overgeneralisation behaviour.

Our data suggests we should reject *Data hypothesis #1* as no significant difference was observed between ATX and PLC in overall memory accuracy for both the explicit and implicit memory tests (Figure 2C-D). In the revised manuscript we provide further evidence to reject *Data hypothesis #1*, using data from the ‘Ring topology task’. Thus, consistent with Figure 2C-D, in the ‘Ring topology task’ no significant group difference was observed for memory of the explicit learned associations (ATX – PLC (n=21:22): $p=0.487$, permutation test) or the implicit contextual associations (ATX – PLC (n=21:22), $p=0.259$, permutation test) (see new Supplementary Figure 4B-C). Again, this suggests that learning under elevated noradrenaline does not quantitatively strengthen memory for all component associations. Instead, as reported in the original manuscript and in the new additional analyses discussed above (see *Reviewer #1*, point #1 and point #2), our data suggests that learning under elevated noradrenaline increases the graded co-activity in the underlying cognitive map, to selectively increase

measures of overgeneralisation behaviour in the ATX relative to the PLC group. We have summarised the above in the revised *Results* and *Discussion* sections.

To gain traction on the precise mechanism that allows for an increase in graded co-activity around the cognitive map, in the revised manuscript we use our ‘*proof-of-concept*’ computational model to explore the following three alternative hypotheses:

Model hypothesis #1: Elevated noradrenaline *during learning* reduces cortical inhibition. As a consequence, surplus excitatory activity generated by new learning is not appropriately rebalanced by local inhibitory connections (I-E). This enables co-activation between cell assemblies, graded by distance, which promotes a gradient of excitatory plasticity between cell assemblies which becomes embedded in the underlying synaptic weights (Figure 4). As a consequence, graded cell assembly overlap becomes embedded in underlying ring structure. Thus, dialing inhibition down *during learning* embeds cognitive maps with ‘leaky’ excitation across memories.

Model hypothesis #2: Local disinhibition *after learning* promotes overgeneralisation. If ‘local’ inhibition is reduced after learning, intra-nodal IE connections fail to counterbalance excitatory co-activity between neighbouring assemblies, leading to graded co-activity in the PLC model (see new Supplementary Figure 9A). We note that this graded co-activity is not as strong as that observed for *Model hypothesis #1*, using the ATX model. In the brain, we speculate that during recall, cell assemblies undergo transient local disinhibition which relieves the effect of local inhibition on excitatory neurons (Letzkus et al., 2015; Ehrlich et al., 2009). However, in our task, participants were not invited to engage in memory recall until day 5, when the effect of drug had washed out. The significant group difference in behaviour on day 5 therefore suggests we should reject *Model hypothesis #2*, as no group difference in local disinhibition is expected at this time point. Instead, ATX but not the PLC group appears to have encoded the underlying cognitive map with graded synaptic weights that drive subsequent measures of overgeneralisation (as shown in Figure 4C).

Model hypothesis #3: Reducing global inhibition after learning promotes overgeneralisation. However, our computational model suggests that graded co-activity across assemblies in the ring is not observed when ‘global’ inhibition is reduced after learning by reducing the strength of all EI connections (to 97%) (see new Supplementary Figure 9B). Further reductions in global inhibition ($\leq 95\%$) make the network unstable with equal co-activity across assemblies in the ring.

Model hypothesis #4: Functional and structural cell assembly overlap can promote graded co-activity in the underlying cognitive map, accounting for overgeneralisation behaviours. Previous computational models elegantly demonstrate this effect in larger networks (Burns et al. 2022). In our computational model, uniform 5% connective sparsity across the network did not account for graded co-activity in the underlying cognitive map (new Supplementary Figure 9C). However, reducing inhibition during learning (in the ATX model) embedded graded cell assembly overlap, where proximal assemblies on the ring structure were more overlapping than distal assemblies. Given that associative learning in the absence of a noradrenergic manipulation can allow for assembly overlap, we also explored the effect of varying the percentage overlap between neighbouring cell assemblies in the PLC network. In the new Supplementary Figure 9D-G, we show that assembly overlap approximately $\geq 40\%$ is required to facilitate a graded co-activity in the underlying cognitive map. Given that no significant behavioural or neural effect of overgeneralisation is observed in the PLC group, we infer that

assembly overlap between associated cues was <40% in the PLC group. Moreover, the reported significant group differences between ATX and PLC suggests a significant difference in assembly overlap embedded at the time of learning. This difference in assembly overlap between ATX and PLC can be attributed to the effect of noradrenaline on inhibitory cell firing which promotes graded co-activity and plasticity across the underlying ring structure (i.e. *Model hypothesis #1*).

To summarise, our proof-of-concept computational model suggests that reducing local but not global inhibition after learning can lead to a spread of association in the underlying neural network (rejecting *Model Hypothesis #3*). These findings are in line with Burns et al. who elegantly demonstrate how the distribution and balance between local (i.e. connected to excitatory assemblies) and global (i.e. connected globally) inhibition can determine the extent to which memory co-activation spreads across an underlying memory structure (Burns et al. 2022). Moreover, given that we observe a significant behavioural difference between ATX and PLC on day 5, after the effect of drug has washed out (Figure 2), our data suggest that the spread-of-association is embedded in the synaptic weights of the cognitive map during learning (rejecting *Model Hypothesis #2*). Given that significant overgeneralisation is not observed in the PLC group, we infer that cell assembly overlap due to associative learning is not sufficient to drive the reported effects of overgeneralisation (rejecting *Model Hypothesis #4*). Rather, the significant spread of association in the ATX group may be attributed to a *drug-induced increase in cell assembly overlap* that occurs due to a decrease in intra-nodal inhibitory rebalancing during learning (*Model Hypothesis #1*). We summarise the above in the revised *Results* and *Discussion*, to further justify the interpretation of the data.

For the reviewer's convenience, we have copied the new Supplementary Figure 9 below.

A Placebo post-learning: weakening local inhibition, via intra-nodal IE

B Placebo post-learning: weakening global inhibition, via all EI

C Baseline assembly overlap, before inhibitory rebalancing

D Placebo: varying % assembly overlap

F Example with 40% assembly overlap

E Placebo: varying % assembly overlap

G Example with 80% assembly overlap

Supplementary Figure 9 | Exploring alternative mechanisms in the neural network model for generating a spread of association in the underlying memory map

A-B The effect of post-learning disinhibition on graded co-activation in the Placebo network. **C-G** The effect of assembly overlap on graded co-activation. **A-C, F-G** Five snapshots of the recurrent spiking neural network. **A-C** Left: Schematic showing the architecture and parameter conditions of the network. Six cell assemblies are pictured as coloured squares. Excitatory and inhibitory connections are drawn in brown and grey, respectively. Dotted lines indicate weaker connections. **A-C, F-G** Middle (*A-C*)/Left (*F-G*): Average firing rate of all excitatory neurons in each assembly, in response to activation of assembly ‘1’ via externally driven input (“stimulus”), in the Placebo network. Right: Average firing

rates of the excitatory neurons in each assembly during the “stimulus” period. **A** Local disinhibition in the post-learning ‘placebo’ network. Local disinhibition is implemented by downregulating intra-assembly inhibitory to excitatory (IE) connections. Activation of cell assembly ‘1’ leads to a graded co-activation across neurons in the other assemblies, relative to their respective distances from assembly ‘1’. While the average firing rates during the stimulus period (right-hand plot) indicate that the qualitative shape of the graded co-activation matches that shown in the ‘atomoxetine’ network shown in Fig 4C (where graded co-activation is observed without local disinhibition), co-activity in the post-learning ‘placebo’ network with local disinhibition is of lower magnitude relative to the post-learning ‘atomoxetine’ network without local disinhibition (‘atomoxetine’: dotted pink; ‘placebo’: dotted blue; without local disinhibition). The associations between nodes weren’t learnt under reduced inhibitory firing in the former case. **B** Global disinhibition in the post-learning ‘placebo’ network. Global disinhibition is implemented by decreasing inhibitory firing across the network by downregulating all excitatory to inhibitory (EI) weights to 97%. Activation of cell assembly ‘1’ does not result in a pronounced graded co-activation across neurons in the other assemblies, relative to their respective distances from assembly ‘1’. When EI weights were further decreased to $\leq 95\%$, all assemblies were equally co-activated with surplus excitation in the network. **C** Baseline structural and functional overlap between assemblies immediately after they are embedded, before they are stabilised by intra-nodal inhibitory rebalancing (“pre-learning” state in Fig 4), and before formation of the ring-structure. Note: in the left-hand schematic, the grey intra-nodal inhibitory connections are absent (compared with Figure 4a). Due to uniform 5% connective sparsity across the network, each assembly is equally connected to all others and equal baseline co-activity is observed across all assemblies in response to driving activity in ‘1’. **D-G** The effect of assembly overlap on graded co-activity in the Placebo network. In the post-learning ‘placebo’ network, activity in assembly ‘1’ together with a varying percentage of neurons in assemblies ‘2’ and ‘6’ are transiently driven for the ‘stimulus’ period. Thus, neurons in neighbouring assemblies are partially driven by the same afferent input, allowing for both functional and structural overlap. We note that assembly overlap can also be achieved using lateral connections or by allocating the same neurons to more than one assembly. In an experimental setting, these different definitions of ‘assembly overlap’ are not necessarily separable. **D-E** Average excitatory current (*D*) and average firing rate (*E*) of all excitatory neurons in assemblies ‘3’ and ‘5’ (i.e. assemblies that are not being directly driven), when varying the percentage overlap between assembly ‘1’ vs. ‘2’ and ‘6’. Co-activity in assemblies ‘3’ and ‘5’ increases steadily when percentage assembly overlap is $\geq 40\%$ and a larger percentage of neurons are co-driven in assemblies ‘2’ and ‘6’. Minimal co-activity is observed when percentage overlap is $< 40\%$. The inflection point of 40% is in part determined by the network size, where a lower percentage would be expected in a larger network. **F-G** Snapshots of the network activity of all excitatory neurons when percentage assembly overlap between assembly ‘1’ and assemblies ‘2’ and ‘6’ is either 40% (*F*) or 80% (*G*). During the “stimulus” period, co-activity can be observed in assemblies ‘3’ and ‘5’, demonstrating a spread of association across the network which is more pronounced in *G* with greater assembly overlap.

Minor issues –

4. It isn’t clear how ‘Overgeneralization errors (%)’ were calculated. The methods indicates that it is the percentage of trials that proximal context is chosen minus the percentage of the trials where any other error was made. Either my understanding of this is wrong or participants are selecting the proximal context even more often than they are selecting the correct context. To unpack – six contexts are shown (16.67% chance), one is correct and this is chosen ~30% of the time (Fig 2), I imagine one of the remaining five is a proximal context (by chance 1 in 5, 20%, of errors should be to a proximal context) while the other four are non-proximal (80% of errors should be this by chance). Fig 2F shows that, even in the for the placebo condition, ‘overgeneralization errors’ are about 0%, which I understand means the proximal context is chosen as often as ‘any other error was made. This means that 50% of the errors, even in the placebo condition are to the proximal context. Given that ~70% of trials were errors, 50% of

those would be 35% of all trials – a value higher than the percent of correct trials. I imagine I’m misunderstanding something – hopefully the authors can spot what and clarify as necessary to avoid such distracting misunderstanding.

We apologise to the reviewer regarding the confusion over our mathematical definition of the overgeneralisation. In the *Implicit memory test* (Figure 2B), we defined ‘*Overgeneralisation errors*’ as the number of times the proximal contextual cue (2 nodes away) was chosen versus any more distal cue (3+ nodes away), correcting for how many proximal versus distal options were presented to the subject on each trial. Notably, by design the proximal cues were represented as options in a fixed proportion of trials: 45% of trials had one proximal cue option and 43% of trials had both proximal cue options. Thus, by accounting for the number of proximal versus distal cues we ensure our behavioural measure is not biased by the number of proximal options presented to the subject each trial. The behavioural data reported in Figure 2F can therefore be interpreted as follows: In the PLC group, the average *Overgeneralisation error* score is close to 0, meaning that participants in this group select the proximal contextual cues at chance level. By contrast, in the ATX group the average *Overgeneralisation error* score is significantly above 0, meaning that participants in this group select the proximal contextual cues above chance level.

In the revised *Methods* section we now include the equation below together with extended explanation. We hope these changes prevent any misunderstanding.

$$\text{Overgeneralisation errors} = \frac{\sum_{\text{trial}=1}^n \left(\frac{\text{proximal choice}_{\text{trial}}}{N \text{ proximal cues}_{\text{trial}}} \right) - \sum_{\text{trial}=1}^n \left(\frac{\text{distal choice}_{\text{trial}}}{N \text{ distal cues}_{\text{trial}}} \right)}{N \text{ trials with opportunity for overgeneralisation error}} \times 100\%$$

5. The timing of the elevated pupil size is extremely slow relative to the task design. The significant difference is only observed a full trial after an oddball.

Thank you for raising this point. Typically, the pupil dilation response to an unexpected stimulus comprises both a phasic and tonic component (e.g. Nassar et al., 2012 Nat. Neurosci; Browning et al., 2015 Nat. Neurosci.; Muller et al., 2019 eLife). The phasic component to surprising stimuli is typically transient, reflecting a deflection from baseline that peaks and returns to baseline within ~3-4 s post stimulus. The tonic component involves a gradual shift in baseline response, typically starting immediately after the phasic peak but lasting for several seconds to minutes (Muller et al., 2019 eLife). These effects can be seen in Figure 3B, where we show the pupil dilation response to surprising stimuli. The expected pattern, where the emergence of the tonic effect is superimposed on the down-swing of the phasic peak, can be seen qualitatively from about 2.5s post stimulus, although the difference between oddball responses in ATX and PLC groups only becomes statistically significant at 6s after the stimulus. In our experience this pattern is typical of pupil dynamics, for example see Browning et al 2015, Nature Neurosci, Figure 4 or Muller et al 2019, eLife figure 3a, right-hand panel.

We take this opportunity to further interpret the phasic and tonic component of the pupil dilation response as a physiological marker of noradrenergic modulation. Both the phasic and tonic components of the pupil dilation response are thought to relate to noradrenergic modulation, where the phasic pupil response has been related to phasic bursts of locus coeruleus (LC) activity and the tonic pupil response has been related to baseline levels of noradrenaline (Aston-Jones & Cohen, 2005; Joshi et al., 2016 Neuron). The ‘adaptive gain

theory' suggests that phasic and tonic components interact according to an inverted-U shape, where increased tonic LC activity may suppress strong phasic responses (Aston-Jones & Cohen, 2005). ATX, a noradrenergic reuptake inhibitor, is reported to increase baseline (tonic) pupil dilation response (Gelbard-Sagiv et al., 2018 *Curr Biology*; Eldar et al., 2013 *Nature Neurosci.*; Orlando et al., 2024 *Brain Comms.*). In our data, reported in Figure 3B, the pupil dilation response may be considered to reflect both baseline LC activity and extracellular levels of noradrenaline, both of which are expected to increase in response to ATX. Consistent with this prediction, and in line with previous studies, 6-10 s after stimulus onset we observe a significant functional increase in the tonic pupil response in the ATX compared to the PLC group. Interestingly, this significant difference in the tonic response is accompanied by a non-significant decrease in phasic pupil response in ATX group relative to PLC, which may be expected due to the inverted-U shape characteristic of phasic relative to the tonic response.

We hope this helps explain why when comparing ATX to PLC we would expect a slow effect in the pupil response to stimulus onset. We have edited the relevant section of the *Results* accordingly to summarise the above text.

6. Fig 1F(top) DABEST representation of data in, using normal distributions, is not well suited to the data which is not normally distributed.

We note that the DABEST plot shows the empirical resampling distribution of the mean proportion correct. The distribution in former Figure 1F looked slightly non-normal, suggesting that bootstrapping is a good approach. However, to avoid confusion we have removed the DABEST plots in Figure 1F.

7. The number of participants and/or degrees of freedom should be listed near where stats are reported.

We apologise for not including this information in the original submission. We have amended the entire manuscript accordingly.

Reviewer #2 (Remarks to the Author):

In this study, Koolschijn et al. investigated how noradrenaline (NA) affects forming implicit associations in a learning task in human volunteers. They used the NA reuptake inhibitor atomoxetine in a double-blind placebo-controlled design. Participants performed a smart associative learning task featuring both an explicit and an implicit learning component. While the explicit learning was unaffected by the treatment, atomoxetine increased over-generalization errors in the implicit domain. Physiological markers of increased NA (GABA tone, pupil size) predicted fMRI signs of this over-generalization effect, which in turn predicted behavioral effects. A simple network model could account for the observed 'spread of associations'. This is an unusually clear model and presentation of a complex and behaviorally relevant concept, with high explanatory power. I only have a few questions and comments.

We'd like to thank the reviewer for their positive and constructive comments.

Major points

1. The correlations between physiological signs of drug effect and fMRI signs of 'spread', as well as between the fMRI and behavioral signatures suggest that there might be a correlation

directly between pupil/GABA tone and over-generalization behavior. Was this tested? If this was feasible to test, it would be interesting to report it.

We would like to thank the reviewer for this point. In the original manuscript we showed that physiological markers of atomoxetine predict fMRI measures of ‘spread-of-association’, which, in turn, predicts behavioural measures of overgeneralisation. In the new Supplementary Figure 7 we report additional correlations between our physiological markers and behaviour.

Following suggestions from *Reviewer #1 Point #1* (see above), we also now include a new, additional behavioural measure of overgeneralisation which we define as the *Mean rank proximity* of the chosen context relative to the correct context in the underlying ring structure, across all error trials in the ‘*Implicit memory*’ test (Figure 3B). Thus, if the *mean rank proximity* score is low, participants are more likely to erroneously pick contextual stimuli that are proximal rather than distal from the target context, suggesting evidence for overgeneralisation. As shown in the new Figure 2G, and consistent with our original definition of overgeneralisation errors shown in Figure 2F, the *Mean rank proximity* score is significantly lower in the ATX group than would be expected by chance ($p < 0.001$, bootstrap test) and significantly lower in the ATX group compared to the PLC group (PLC - ATX: $p = 0.031$, permutation test). In light of your comment above, in the revised manuscript we assess the relationship between our physiological markers and both the original behavioural measure, namely *Overgeneralisation errors* (Figure 2F), and this new behavioural measure, namely *Mean rank proximity* (Figure 2G).

No significant correlations were observed in either group, with results summarised below and presented in the new Supplementary Fig. 7. Thus, while a single dose of atomoxetine induced expected changes in physiological markers of noradrenergic arousal (namely increased pupil response and reduced cortical inhibition), these physiological markers provide a precise predictor for neural and, in turn, behavioural measures of overgeneralisation. Our findings demonstrate that a measure of the cognitive neural response to atomoxetine, and not mere physiological response to the drug, is necessary to predict overgeneralisation errors in behaviour. Notably, these findings mirror those observed in our previous work, where we used non-invasive brain stimulation (tDCS) to induce variability in cortical excitability (Koolschijn et al., Neuron 2019). There, as observed here, we found that a neural measure of cortical excitability, and not mere application of tDCS, were necessary to predict behavioural measures of memory interference.

We summarise the above in the revised *Discussion* and have copied the new Supplementary Figure 7 below, for the reviewer’s convenience.

ATX:

Overgeneralisation errors α GABA+ $n=19$; $r_{17}=-0.135$; $p=0.581$;

Mean rank proximity α GABA+ $n=19$; $r_{17}=0.158$; $p=0.571$;

Overgeneralisation errors α pupil response: $n=17$; $r_{15}=0.277$; $p=0.281$.

Mean rank proximity α pupil response: $n=17$; $r_{15}=-0.282$; $p=0.272$.

PLC:

Overgeneralisation errors α GABA+ $n=18$; $r_{16}=-0.267$; $p=0.282$;

Mean rank proximity α GABA+ $n=18$; $r_{16}=-0.131$; $p=0.603$;

Overgeneralisation errors α pupil response: $n=17$; $r_{15}=-0.034$; $p=0.898$.

Mean rank proximity α pupil response: $n=17$; $r_{15}=0.232$; $p=0.371$).

Supplementary Figure 7 | Relationship between physiological measures of elevated noradrenaline and behavioural measures of overgeneralisation

In the ATX group, a single dose of atomoxetine induced expected changes in physiological markers of increased noradrenaline (namely increased pupil response and reduced cortical inhibition, Fig. 3B,E). These physiological markers provide a precise predictor for neural (Fig. 6E-F) and, in turn, behavioural measures of overgeneralisation (Fig. 6A-D). By assessing the relationship between physiological measures of elevated noradrenaline and two behavioural measures of overgeneralisation (namely, *Overgeneralisation errors* and *Mean rank proximity*, Figure 2F-G respectively, see *Methods*), here we show that a measure of the neural response to atomoxetine, and not mere physiological response to the drug, is necessary to predict overgeneralisation errors in behaviour. **A** No significant relationship was observed between GABA+ in LOC (Fig. 3E) and *Overgeneralisation errors* (Fig. 2F) in either the ATX or PLC group (ATX: $n=19$, Spearman correlation: $r_{17} = -0.135$, $p=0.581$; PLC: $n=18$, Spearman correlation: $r_{16} = -0.267$, $p=0.282$). **B** No significant relationship was observed between GABA+ in LOC (Fig. 3E) and *Mean rank proximity* (Fig. 2G) in either the ATX or PLC group (ATX: $n=19$, Spearman correlation: $r_{17} = 0.158$, $p=0.517$; PLC: $n=18$, Spearman correlation: $r_{16} = -0.131$, $p=0.603$). **C** No significant relationship was observed between pupil dilation response (Fig. 3B) and *Overgeneralisation errors* (Fig. 2F) in either the ATX or PLC group (ATX: $n=17$, Spearman correlation: $r_{15} = 0.277$, $p=0.281$; PLC: $n=17$, Spearman correlation: $r_{15} = -0.034$, $p=0.898$). **D** No significant relationship was observed between pupil dilation response (Fig. 3B) and *Mean rank proximity* (Fig. 2G) in either the ATX or PLC group (ATX: $n=17$, Spearman correlation: $r_{15} = -0.282$, $p=0.272$; PLC: $n=17$, Spearman correlation: $r_{15} = 0.232$, $p=0.371$).

2. It would be worthwhile to elaborate on the rationale behind focusing on LOC (is that ‘lateral occipital complex’?). Was a change in GABA tone not expected, as well as observed elsewhere?

Thanks for this comment. Atomoxetine has a systemic effect and therefore a reduction in GABA would be expected across the whole brain. Due to time constraints, it was not possible

to acquire MRS data in more than two voxels as data acquisition took around 15 minutes per voxel.

The rationale for acquiring MRS data from LOC was motivated by the nature of the stimuli in our task (bird stimuli positioned in scene images) combined with evidence that LOC is strongly implicated in object processing, by contributing to basic object recognition and scene perception while processing object shape (Grill-Spektor et al., 1999; Grill-Spektor et al., 2001).

In the new Supplementary Figure 5 we now show that indeed LOC shows an increase in BOLD signal in response to task stimuli, indexed through fMRI repetition suppression. In LOC, we therefore expected any change in glu/GABA in response to ATX to result in task relevant neural changes. In addition, signal-to-noise (SNR) in MRS data acquired from LOC is typically high, allowing for robust and reliable measures. Indeed, we show that temporal SNR (tSNR) is significantly higher in LOC compared to V1, the location of our second MRS voxel (Supplementary Figure 6). This significant difference in tSNR between LOC and V1 may explain why we observe no significant change in GABA+ in our second voxel, namely V1. By contrast, other task relevant brain regions, such as hippocampus, are not well suited for MRS acquisition due to their proximity to air filled cavities and due to decreased MRS signal strength and thus lower SNR in deeper brain regions. During piloting for this study, we tried to acquire MRS data from hippocampus, but aborted this approach as data quality was very poor.

The above is now summarised in the revised *Results*. For the reviewer's convenience we have copied the relevant part of Supplementary Figure 5 below.

Supplementary Figure 5 (cropped)

A ROI consisting of two 10mm radius spheres, where the right sphere was centered on the group-average MRS voxel positioned in LOC (Figure 3C), and the left sphere being mirrored along the x-axis. Used for small volume correction (SVC) for the bird stimuli repetition suppression contrast. **B** Overlap of MRS voxel positioned in LOC (Figure 3C) and T-statistic map for fMRI repetition suppression for the bird stimuli. A significant repetition suppression effect was observed in LOC across all participants (SVC with ROI shown in *A*, $t_{42}=4.23$, $p=0.010$, MNI coordinates, Supplementary Data Table 1). The repetition suppression contrast was generated with a parametric regressor indicating the number of trials since the same stimulus was last presented (see *Methods*).

3. The Authors tend to report direct statistical comparisons of the groups, which is the better approach than saying 'significant in A but not significant in B'. I did not find this for the correlations of the physiological markers and 'neural spread of associations' (lines 405-407, 417-419). Was there a significant difference in the direct comparisons? In this regard, the sample sizes are also of relevance; since there were some exclusions according to the Methods, it would make sense to provide all actual sample sizes for the statistical comparisons in the figure legends.

We apologise for not including these in the original draft. The between group comparison for the correlation between the pupil dilation response and cross-stimulus suppression was statistically significant in left and right hippocampus (left: $t_{19}=4.13$, $p=0.026$; right: $t_{19}=3.89$; $p=0.042$) and in the left parahippocampus ($t_{19}=3.48$, $p=0.046$). No significant difference was observed for a between group comparison for the correlation between GABA and cross-stimulus suppression. These between group comparisons are now included in the revised *Results* section, shown in the new Supplementary Fig. 11 and reported in Supplementary Table 14.

Furthermore, following your recommendation and in response to Reviewer #1 minor p4, we now report the sample size for each statistical comparison in the figure legends. Thank you for raising this point.

Supplementary Figure 11 (cropped)

H No significant group difference (ATX vs. PLC) was observed for the relationship between GABA⁺ in LOC and the neural spread of association effect in hippocampus-parahippocampus (ATX – PLC (**n=19:17**): SVC with parahippocampus-hippocampus ROI, $t_{30}=3.17$, $p=0.195$, MNI coordinates). **I** In the PLC group, no significant correlation was observed between the pupil dilation response to surprising stimuli and the neural spread of association effect (SVC with parahippocampus-hippocampus ROI, $n=12$, $t_8=3.63$, $p=0.501$ MNI coordinates). **J** A trend towards a significant group difference (ATX vs. PLC) was observed for the relationship between the pupil dilation and the neural spread of association in parahippocampus-hippocampus (ATX – PLC (**n=15:12**): SVC with anatomical parahippocampus-hippocampus ROI, $t_{19}=3.45$, $p=0.078$, MNI coordinates, Supplementary Table 12). **Post hoc tests revealed significant group differences in left and right hippocampus and left parahippocampus (ATX – PLC (n=15:12): SVC with left hippocampus ROI, $t_{19}=4.13$, $p=0.026$; SVC with right hippocampus ROI, $t_{19}=3.89$, $p=0.042$; SVC with left parahippocampus ROI, $t_{19}=3.48$, $p=0.046$).**

4. I found the correlation results pertaining to Figure 5 very exciting. At the same time, I was somewhat confused by the figure. What exactly is plotted in the scatter plots in Fig.5E, G and

H? Is every dot representing a single participant? If so, why are the plots ‘illustrative’ – is this a selection of the data? If so, it would be better to present all data in the figure.

We apologise to the reviewer regarding the confusion with labelling the scatter plots in Figure 5 ‘illustrative’. We include these scatter plots to provide an intuitive display of the results shown in the fMRI contrasts. These scatter plots include all participants, and each dot is indeed a single participant. We previously labelled these scatter plots as illustrative because the significance of the effect is already tested (with statistical correction) and reported for the fMRI contrast. As such, the scatter plot does not represent a separate result to the fMRI contrast. We have now reworded the accompanying figure legend to reflect this and hope these changes provide clarification.

5. The Methods state that participants were trained to criterion. Did learning time differ in the two groups? Did learning time show any correlation with the results (behavioral, neural, physiological signs)?

Thank you for raising this important point. In the new Supplementary Figure 12A we now show that there is no significant difference in learning time between the ATX and PLC groups (ATX – PLC (n=22:22): permutation test $p=0.409$). Furthermore, we observed no significant correlation between learning time and our neural measure for spread of association and physiological measures of noradrenaline, for either the ATX group (Spread of association, XSS: $r_{20}=-0.321$, $p=0.156$; GABA+: $r_{17}=0.270$, $p=0.262$; Pupil response: $r_{15}=0.029$, $p=0.913$) or the PLC group (Spread of association, XSS: $r_{19}=-0.368$, $p=0.102$; Pupil response $r_{15}=0.069$, $p=0.795$), with the exception of learning time and GABA+ in the PLC group ($r_{16}=-0.501$, $p=0.036$). We speculate that the latter may be explained by the fact that participants who completed the learning in less time would have had to wait longer for their MRI scan, and therefore may have been more relaxed resulting in higher GABA (e.g. Abdou et al., 2006).

When assessing the relationship between learning time and our behavioural measures of spread of association, we observed no significant correlation in the ATX group (*Overgeneralisation errors*: $r_{20}=0.038$, $p=0.865$; *Mean rank proximity*: $r_{20}=0.025$, $p=0.912$). In the PLC group a significant correlation was observed between learning time and *Overgeneralisation errors* ($r_{20}=0.525$, $p=0.013$) with a similar trend for *Mean rank proximity* ($r_{20}=-0.366$, $p=0.094$), which may potentially be explained by an increase in assembly overlap across associated cues when learning occurs across a longer time interval (see *Model hypothesis #4* in response to *Review #1 point #3*).

We summarise the above in the revised Results section and in the new Supplementary Figure 12, which we copy below for the reviewer’s convenience.

Supplementary Figure 12 | Learning time did not differ between groups and did not relate to behavioural, neural or physiological measures in the ATX group

A On Day 1, participants learned the stimulus associations in 29.72 minutes on average. There was no significant difference in learning time between groups (ATX – PLC ($n=22:22$): permutation test $p=0.409$). Upper: Bootstrap-coupled estimation (DABEST) plots. Black dot, mean; black ticks, 95% confidence interval; filled curve, sampling error distribution. Lower left: memory accuracy (mean \pm SEM). Lower right: null distribution of the group differences generated by permuting subject labels, purple dot: true group difference. **B** In the ATX group, no relationship was observed between learning time and overgeneralisation errors (Fig. 2F) (ATX: $n=22$, Spearman correlation: $r_{20}=0.038$, $p=0.865$). In the PLC group, learning time positively predicted overgeneralisation errors (PLC: $n=22$, Spearman correlation: $r_{20}=0.525$, $p=0.013$). **C** No relationship was observed between learning time and mean error proximity rank (Fig. 2G) in either the ATX or PLC group (ATX: $n=22$, Spearman correlation: $r_{20}=0.024$, $p=0.916$; PLC: $n=22$, Spearman correlation: $r_{20}=-0.366$, $p=0.094$). **D** No relationship was observed between learning time and the XSS regressor for neural spread of association (Fig. 5C, Supplementary Fig. 11B) in either the ATX or PLC group (ATX: $n=22$, Spearman correlation: $r_{20}=-0.321$, $p=0.156$; PLC: $n=21$, Spearman correlation: $r_{19}=-0.368$, $p=0.102$). **E** In the ATX group, no relationship was observed between learning time and GABA+ in LOC (Fig. 3E) (ATX: $n=19$, Spearman correlation: $r_{17}=0.270$, $p=0.262$). In the PLC group, learning time positively predicted GABA+ in LOC (PLC: $n=18$, Spearman correlation: $r_{16}=-0.501$, $p=0.036$). **F** No relationship was observed between learning time and pupil dilation response (Fig. 3B) in either the ATX or PLC group (ATX: $n=17$, Spearman correlation: $r_{15}=0.029$, $p=0.913$; PLC: $n=17$, Spearman correlation: $r_{15}=0.069$, $p=0.795$).

Minor points

1. Figure legend for panel 1F: subpanel assignments (e.g. upper right) do not seem to fit the image.

We thank the reviewer for flagging this error. We have corrected the legend of Figure 1F in the revised manuscript and note that Figure 1F now also includes fewer subpanels in response to *Reviewer #1, minor point #3*.

2. 'Runaway excitation' suggests an uncontrolled processes: I suggest rephrasing this.

Thanks for this suggestion - we have rephrased accordingly.

Disclaimer: Not having performed those methods myself, I am not in a good position to judge the methodology for the MRS, and I could only assess the broad approaches with respect to the fMRI methodologies.

I believe that openness and transparency increase the fairness of peer review. Therefore, I am signing my reviews.

Balazs Hangya

Reviewer #3 (Remarks to the Author):

In this paper, the authors studied the effect of noradrenaline on the association of memory items in the formation of the hippocampal cognitive map of human patients. To this end, the authors designed a behavioral task in which the patients learn the association between bird-bird pairs and contextual cues. Presentations of these images had a virtual ring structure, "- bird - cue - bird - cue -..." The authors showed that an increasing level of noradrenaline causes the overgeneralization of the bird-bird association over a long distance along the ring. The authors constructed a computational network model to suggest that the overgeneralization is caused by the reduction of inhibitory plasticity by the noradrenergic level increase.

The experimental paradigm seems to be novel, and the topic is interesting. However, I am unsure whether the results shown here are qualitatively and quantitatively sufficient to warrant publication in Nat Comm. The quantitative information about the overgeneralization is also limited.

We would like to thank the reviewer for their constructive comments. We are pleased to see that the reviewer considers the experimental paradigm to be novel. In the revised manuscript we now provide new quantitative evidence (both behavioural and neural) to demonstrate that learning under elevated noradrenaline increases the spread of association around the underlying cognitive map.

Specifically, we now include two new behavioural measures of overgeneralisation, in each case showing significant evidence for overgeneralisation in the ATX compared to the PLC group (see new Figure 2G and new Supplementary Figure 4). At the neural level we now fit the model to the data to demonstrate a second quantitative measure of spread of association in the neural

data (see new Figure 6B). In response to your comments below, we have also taken the opportunity to use our computational model to explore how several different mechanisms can give rise to a spread of association (see new Supplementary Figure 9). We hope the reviewer agrees that these changes have enriched and strengthened the manuscript.

As the reviewer's comments all relate to the computational modelling, we would also like to take this opportunity to emphasise that the model is a proof-of-concept model, designed to *aid interpretability of the data* by providing some mechanistic insight. Despite incorporating several biologically plausible features, the model is grossly simplified relative to a biological system. Therefore, while the modelling provides a useful test-bed, our empirical findings alone reveal a new understanding of the role of noradrenaline. The empirical data we report are both qualitatively and quantitatively novel. Indeed, we present an unusual and rare approach that combines a pharmacological manipulation with fMRI, MRS and rich behavioural analyses, enabled by our carefully designed cognitive task. We hope the reviewer agrees that our findings will be of significant interest to a broad neuroscience audience, while also carrying translational value for neuropsychiatric conditions such as Post-Traumatic Stress Disorder (PTSD) where elevated levels of noradrenaline persist after a traumatic event.

Below we address each of your concerns in turn.

Major concerns:

Several studies have shown that memory engrams encoding sensory and behavioral experiences that occur in a temporal vicinity tend to overlap (e.g., Mitashita et al., Nature 1988; Cai et al., Nature 2016). In addition, several models have been proposed to account for the overlapping cell assemblies (e.g., Griniasty et al., Neural Comput 1993; Haga and Fukai, Phys Rev Lett, 2019; Burns et al., eNeuro 2022). The authors, however, simply assumed that cell assemblies encoding memory items are orthogonal and mutually non-overlapping. This setting of the model looks over-simplified. What experimental data justify this assumption?

We would like to apologise for not making this more clear in the previous version of the paper. We agree with the reviewer that some degree of both **functional and structural** overlap is expected across neighbouring cell assemblies due to associative learning of bird-bird pairs, in line with empirical evidence (e.g. Cai et al., 2016; Miyashita et al., 1988) and other computational models (e.g. Griniasty et al., 1993; Haga and Fukai, 2019; Burns et al., 2022). In the context of our model we can define "overlap" in three ways that are not necessarily separable in an experimental setting. A) Two assemblies are connected laterally, such that a stimulus delivered exclusively to assembly 1 will cause a response in assembly 2 by way of lateral excitation (strictly functional overlap). B) Two assemblies 1 and 2 could be partially driven by the same afferent input such that some neurons in assembly 2 will respond to the stimulus that is mainly delivered to assembly 1 (functional and structural overlap). C) Assemblies 1 and 2 share some of the same neurons such that a stimulus to assembly 1 will lead to some co-activation of assembly 2 by the nature of these shared neurons (functional and structural overlap). Cases B and C cannot be easily disambiguated as we define assemblies by the input they represent. In the previous version of our paper we explored only option A), overlap by way of lateral activation of neighboring assemblies (albeit without discussing this overlap sufficiently, which we have now amended). In response to your comment, we have now also explored option B/C, to understand if *cell assembly overlap of any kind, due to associative learning, is sufficient to explain overgeneralisation as observed in ATX but not PLC*.

First, we note that in our computational model, uniform 5% connective sparsity was implemented across the network, accounting for structural overlap between the 6 cell assemblies. This also leads to functional overlap, i.e., co-activation, before intra-nodal inhibitory rebalancing takes effect. We now discuss this and show it more clearly in the new Supplementary Fig. 9C. Once inhibitory rebalancing has taken effect, functional assembly overlap is no longer observed (Fig. 4A), demonstrating how the precise balance between excitation and inhibition controls the relative co-activation between assemblies.

After associative learning in the model, we would expect some degree of functional and structural overlap between neighbouring cell assemblies due to excitatory plasticity, in line with previous experimental data (e.g. Cai et al., 2016; Miyashita et al., 1988).

To explore the relationship between functional assembly overlap and overgeneralisation, in our PLC model we varied the degree of assembly overlap, in line with option B). We note that our model was not designed to capture the full complexity of a biological system and decisions were made to simplify various features. In the new Supplementary Fig. 9D-G we therefore test evidence for overgeneralisation while varying the percentage overlap between neighbouring assemblies. We show that in the PLC model overgeneralisation is only observed when neighbouring assemblies overlap by $\geq 40\%$, a value in-part determined by the network size. However, given that we don't observe significant evidence for overgeneralisation errors in our PLC participants, we conclude that in the brain, assembly overlap due to associative learning is not sufficient to drive overgeneralisation in behaviour. However, in the ATX group we do observe significant overgeneralisation together with significant between group differences (ATX - PLC). This raises the possibility that elevated noradrenaline during learning increases cell assembly overlap. Indeed, this is precisely what we show functionally in Figure 4C, where graded co-activity across the network results in graded synaptic plasticity, increasing overlap between cell assemblies in a graded manner. In the revised manuscript we now quantify this assembly overlap in the accompanying figure legend (Figure 4C), as follows:

‘The reduced firing in inhibitory neurons promoted a graded change in synaptic weights, embedding the spread of association in the memory map to outlast the effect of atomoxetine itself (lower right). This resulted in overlap between cell assemblies. For example, assembly ‘1’ now overlapped with each of the other assemblies by the following percentages: ‘2’: 54%; ‘3’: 31%. ‘4’: 19%, ‘5’: 28%, ‘6’: 57%.’

Finally, it is interesting to compare our results with recent empirical data in rats demonstrating an increase in the specificity of contextual fear memory when noradrenaline is infused into basal-lateral amygdala, but not when noradrenaline is infused into central amygdala or hippocampus (Atucha et al., 2025). The effect of noradrenaline on cell assembly overlap therefore appears to vary across brain regions and depends on the type of memory in question. Our empirical data and computational model together suggest that when learning a cognitive map under elevated noradrenaline, cell assembly overlap increases in the hippocampal representation of the map, due to elevated between-assembly plasticity facilitated by a decrease in inhibitory cell firing.

We summarise the above in the revised *Results* and *Discussion* sections. For the reviewer's convenience we have copied the new Supplementary Figure 9 below, which also includes exploration of the effect of local and global disinhibition on overgeneralisation (further inspired by Burns et al., 2022).

A Placebo post-learning: weakening local inhibition, via intra-nodal IE

B Placebo post-learning: weakening global inhibition, via all EI

C Baseline assembly overlap, before inhibitory rebalancing

D Placebo: varying % assembly overlap

F Example with 40% assembly overlap

E Placebo: varying % assembly overlap

G Example with 80% assembly overlap

Supplementary Figure 9 | Exploring alternative mechanisms in the neural network model for generating a spread of association in the underlying memory map

A-B The effect of post-learning disinhibition on graded co-activation in the Placebo network. **C-G** The effect of assembly overlap on graded co-activation. **A-C, F-G** Five snapshots of the recurrent spiking neural network. **A-C, F-G** Left: Schematic showing the architecture and parameter conditions of the network. Six cell assemblies are pictured as coloured squares. Excitatory and inhibitory connections are drawn in brown and grey, respectively. Dotted lines indicate weaker connections. **A-C, F-G** Middle (*A-C*)/Left (*F-G*): Average firing rate of all excitatory neurons in each assembly, in response to activation of assembly ‘1’ via externally driven input (“stimulus”), in the Placebo network. Right: Average firing

rates of the excitatory neurons in each assembly during the “stimulus” period. **A** Local disinhibition in the post-learning ‘placebo’ network. Local disinhibition is implemented by downregulating intra-assembly inhibitory to excitatory (IE) connections. Activation of cell assembly ‘1’ leads to a graded co-activation across neurons in the other assemblies, relative to their respective distances from assembly ‘1’. While the average firing rates during the stimulus period (right-hand plot) indicate that the qualitative shape of the graded co-activation matches that shown in the ‘atomoxetine’ network shown in Fig 4C (where graded co-activation is observed without local disinhibition), co-activity in the post-learning ‘placebo’ network with local disinhibition is of lower magnitude relative to the post-learning ‘atomoxetine’ network without local disinhibition (‘atomoxetine’: dotted pink; ‘placebo’: dotted blue; without local disinhibition). The associations between nodes weren’t learnt under reduced inhibitory firing in the former case. **B** Global disinhibition in the post-learning ‘placebo’ network. Global disinhibition is implemented by decreasing inhibitory firing across the network by downregulating all excitatory to inhibitory (EI) weights to 97%. Activation of cell assembly ‘1’ does not result in a pronounced graded co-activation across neurons in the other assemblies, relative to their respective distances from assembly ‘1’. When EI weights were further decreased to $\leq 95\%$, all assemblies were equally co-activated with surplus excitation in the network. **C** Baseline structural and functional overlap between assemblies immediately after they are embedded, before they are stabilised by intra-nodal inhibitory rebalancing (“pre-learning” state in Fig 4), and before formation of the ring-structure. Note: in the left-hand schematic, the grey intra-nodal inhibitory connections are absent (compared with Figure 4a). Due to uniform 5% connective sparsity across the network, each assembly is equally connected to all others and equal baseline co-activity is observed across all assemblies in response to driving activity in ‘1’. **D-G** The effect of assembly overlap on graded co-activity in the Placebo network. In the post-learning ‘placebo’ network, activity in assembly ‘1’ together with a varying percentage of neurons in assemblies ‘2’ and ‘6’ are transiently driven for the ‘stimulus’ period. Thus, neurons in neighbouring assemblies are partially driven by the same afferent input, allowing for both functional and structural overlap. We note that assembly overlap can also be achieved using lateral connections or by allocating the same neurons to more than one assembly. In an experimental setting, these different definitions of ‘assembly overlap’ are not necessarily separable. **D-E** Average excitatory current (*D*) and average firing rate (*E*) of all excitatory neurons in assemblies ‘3’ and ‘5’ (i.e. assemblies that are not being directly driven), when varying the percentage overlap between assembly ‘1’ vs. ‘2’ and ‘6’. Co-activity in assemblies ‘3’ and ‘5’ increases steadily when percentage assembly overlap is $\geq 40\%$ and a larger percentage of neurons are co-driven in assemblies ‘2’ and ‘6’. Minimal co-activity is observed when percentage overlap is $< 40\%$. The inflection point of 40% is in part determined by the network size, where a lower percentage would be expected in a larger network. **F-G** Snapshots of the network activity of all excitatory neurons when percentage assembly overlap between assembly ‘1’ and assemblies ‘2’ and ‘6’ is either 40% (*F*) or 80% (*G*). During the “stimulus” period, co-activity can be observed in assemblies ‘3’ and ‘5’, demonstrating a spread of association across the network which is more pronounced in *G* with greater assembly overlap.

Related to the above comment, there are no quantitative comparisons between the model and activity data (e.g., in Fig. 4), which also limits the validity of the proposed model and inhibitory mechanism of overgeneralization.

Thank you for this suggestion. As mentioned above, our proof-of-concept computational model is grossly simplified relative to a biological system. One feature of the model that we chose to simplify was the size of the embedded ring topology, using only 6 assemblies in the model compared to 11 nodes for participants. This simplification precluded direct comparison between the data and the model. Nevertheless, in the revised manuscript we make direct quantitative comparison between the model and a subset of the data, as detailed below.

In the original fMRI analysis presented in Figure 5 (now Figure 5C) the parametric regressor used to test evidence for cross-stimulus suppression (Figure 5B) was constructed using a z-scored exponential function ($1-2[\text{link distance}]$) to account for the predicted spread of association by link distance. In the revised manuscript, we now construct a second parametric regressor from model outputs, to allow direct comparison between the model and data. To construct this second parametric regressor, we used the peak excitatory firing rates of assemblies in the model after the associations between cell assemblies have been embedded and plasticity rules given time to apply (Figure 4C, middle panel, ATX group). Notably, given the model only included 6 assemblies, this second parametric regressor was used to predict cross-stimulus suppression across only a subset of trials where the between-node distance ranged from 1 to 3 steps. In the right hippocampus, the model derived parametric regressor revealed significant evidence for spread of association in the ATX group, together with a significant between group difference (ATX - PLC). For the reviewer's convenience, this new result is copied below.

Figure 5 (cropped)

G-I Quantitative comparison between the ATX neural network model (Figure 4C) and data. XSS for spread of association in the ATX group was modelled using the peak firing rate of assemblies in the model (Figure 4C, middle) as a predictor for co-activation across link distance 1-3. **G** In the ATX group a significant spread of association was observed in right hippocampus (ATX: $n=22$, $p=0.024$, effect size computed from 10,000 bias-corrected bootstrapped resamples²⁷), with a trend in right parahippocampus (ATX: $n=22$, $p=0.055$, bootstrap test). **H** T-statistic map showing a trend towards a group difference (ATX-PLC) for the spread of association effect (SVC with parahippocampus-hippocampus ROI: ATX - PLC ($n=22:21$), $t_{41}=2.96$, $p=0.136$, MNI coordinates, Supplementary Table 5). **I** Post hoc tests revealed a significant group difference (ATX-PLC) in right hippocampus for the spread of association effect (ATX - PLC ($n=22:21$): $p=0.041$, permutation test). Lower: individual β parameter estimates obtained from the right hippocampus (mean \pm SEM). Upper: DABEST plots. Black dot, mean; black ticks, 95% confidence interval; filled curve, sampling error distribution.

Is heterosynaptic plasticity in Eq. 7 crucial for the bird-bird association? How is overgeneralization affected by the heterosynaptic plasticity?

In general, regardless of the exact identity of the underlying mechanism, plasticity mechanisms must meet two requirements to enable stable memory function. First, unstable (Hebbian) forms of plasticity are complemented with compensatory (non-Hebbian) forms of plasticity acting on the same timescale. Second, to create stable fixed points in long-term plasticity dynamics, compensatory plasticity mechanisms must exist that engage at high postsynaptic activity (Zenke et al., 2015). Heterosynaptic plasticity contributes to both these two requirements,

providing a compensatory (non-Hebbian) form of plasticity that engages at high postsynaptic activity to create a stable fixed point in the plasticity dynamics.

In our proof-of-concept computational model we implemented a framework that formalises co-dependent plasticity based on multiple empirical findings (Agnes and Vogels, 2024). This co-dependent plasticity framework includes a heterosynaptic plasticity term to counterbalance the destabilising effects of excitatory synaptic potentiation. In the new Supplementary Figure 10 (copied below) we demonstrate that without the heterosynaptic plasticity term the excitatory synaptic weights do not stabilise and reach their maximum possible value, with excitatory firing attaining implausible firing rates. Therefore, in our model, heterosynaptic plasticity functionally contributes to network stability in both the PLC and ATX networks, even prior to assembly embedding. In the ATX network, there is an additional stabilising effect of heterosynaptic plasticity that counteracts the surplus excitation in the network due to learning under reduced inhibitory firing.

In addition to the qualitative description of heterosynaptic plasticity provided above, we can mathematically formalise the contribution of heterosynaptic plasticity as follows:

The learning rule of the j^{th} excitatory synapse onto the postsynaptic neuron is defined as:

$$\begin{aligned} \frac{dw_j(t)}{dt} &= \phi_E(E_j(t), I(t); S_j(t), S_{\text{post}}(t)) \\ &= [A_{\text{LTP}} x_j^+(t) S_{\text{post}}(t) E_j(t) - A_{\text{LTD}} y_{\text{post}}^-(t) S_j(t) w_j(t) \\ &\quad - A_{\text{het}} y_{\text{post}}^E(t) S_{\text{post}}(t) (E_j(t))^2] \exp\left[-\left(\frac{I(t)}{I_s}\right)^\gamma\right] \end{aligned} \quad (7)$$

Which includes both LTP and LTD components. Importantly, the total excitatory input “E” to a synapse controls Hebbian LTP. Without heterosynaptic plasticity, this homosynaptic Hebbian LTP allows excitatory synaptic weights to constantly increase (as shown in the new Supplementary Figure 10B). As illustrated by the schematic below, homosynaptic Hebbian LTP (green) and heterosynaptic plasticity (orange) together formulate a common setpoint (red) for total excitatory input, which prevents runaway excitation. Hebbian LTD, on the other hand, is dependent only on synapse-specific spike-time dependent activity, and hence cannot prevent runaway excitation induced by total excitatory input “E”.

Courtesy: Agnes and Vogels (2024)

To summarise, in our model heterosynaptic plasticity provides a stabilising mechanism to counteract the excitatory plasticity that enables embedding of sensory representations (i.e. cell assemblies; ATX and PLC), embedding of associative memories (i.e. ring topology; ATX and PLC), and the graded excitatory synaptic weights across assemblies (ATX model only). Therefore, heterosynaptic plasticity can be considered to prevent destabilising run-away excitation to effectively increase the capacity of the network for information storage.

In the revised manuscript we summarise the above in the revised *Methods* section. For the reviewer's convenience we have copied the new Supplementary Figure 10 below.

Supplementary Figure 10 | Neural network activity showing the effect of heterosynaptic plasticity in stabilising the network via codependent plasticity rules

Our neural network model employs a "codependent" plasticity rule⁴⁷ which includes both LTP and LTD components. While the total excitation ("E") to a synapse controls Hebbian LTP, without heterosynaptic plasticity this homosynaptic Hebbian LTP allows excitatory synaptic weights to constantly increase. **A-B** Two snapshots of network behaviour, with (*A*) and without (*B*) heterosynaptic plasticity during the initial burn-in stabilisation period of the network, before assemblies are embedded. Network activity is driven only by background activity as described in the *Network and Simulation* section of the *Methods*. Top: Average firing rates through time of six random groups of excitatory neurons, where each group is composed of 100 neurons. Middle: Synaptic weights through time of 500 randomly selected excitatory synapses (w_{EE}) from the network. Bottom: Average net excitatory current (implemented in eq. 4, eq. 5, eq. 7 of *Methods*), shown through time for 100 randomly selected excitatory neurons in the network. **A** With heterosynaptic plasticity, the network stabilises after ~20 seconds of background activity with reasonable firing rates, excitatory weights and excitatory currents, thus establishing a balanced recurrent spiking neural network. **B** Without heterosynaptic plasticity, the excitatory-to-excitatory weights (w_{EE}) are not sufficiently stabilised. As a consequence, w_{EE} reach their maximum possible value (1nS), leading to increasing exponential excitatory currents in the network, and, in turn, implausible excitatory firing rates are attained in the network.

The explanation of network simulations is confusing. How E-E and I-E connections were modified or fixed in the simulations is hard to understand. For instance, in the Inter-nodai Association, the authors mention w_{EE} was fixed at the same values, but w_{EI} was set at different values in the placebo and atomoxetine conditions. Does this imply that the excitatory and inhibitory plasticity rules were not applied to these synapses? Then, which synapses were modifiable?

We would like to thank the reviewer for raising this point of confusion. We apologise that the relevant *Methods* section of the submitted manuscript was not sufficiently clear. Below, we provide more detail on how the codependent plasticity rules were applied in our neural network models. We have also revised the *Methods* section accordingly, to make these points clear.

Throughout our simulations we implemented a co-dependent synaptic plasticity rule, following excitatory and inhibitory plasticity rules enlisted in eqns. 7-13. For simplicity, only the excitatory synapses projecting onto excitatory cells (w_{EE}) and the inhibitory synapses projecting onto excitatory cells (w_{IE}) were plastic. The remaining synapses (w_{EI} and w_{II}) were held static after their values were initialised. We embedded six cell assemblies into the network by connecting independent Poisson input groups (representing external stimuli) to distinct sets of 100 excitatory neurons each which formed representations of the six stimuli (1:6). To achieve E:I balance on these six assemblies, for 25 inhibitory neurons per assembly the local inhibitory input (w_{IE}) was manually strengthened, rather than relying on plasticity to reach this balance dynamically. This allowed us to stably form well-defined cell assemblies with distinct excitatory and inhibitory neurons, enabling clear identification of the specific local inhibitory interneurons associated with each assembly.

The functional role of IE connections that are plastic under the codependent plasticity rule becomes critical later in the simulations, when we introduce external perturbations to the network. To simulate learning under atomoxetine, we manipulated inhibitory synaptic weights to mimic learning during a period of reduced inhibitory firing. Thus, in the atomoxetine neural network simulation, we set all excitatory synapses projecting onto inhibitory cells (w_{EI}) to 97% the weight used for the placebo neural network, namely $w_{EI} = 0.3011$ nS. As above, w_{EI} subsequently remained static throughout the simulations.

In both the atomoxetine and placebo neural networks we introduced the ring topology by manually strengthening the connections between neighbouring cell assemblies, by increasing the weight of excitatory synapses projecting onto excitatory cells (w_{EE}) to the same value in both placebo and atomoxetine neural networks. To prevent runaway excitation, this increase in E-E potentiation was counterbalanced by manual strengthening of inhibitory connections between neighbouring cell assemblies, by increasing the weight of inhibitory synapses projecting onto excitatory cells (w_{IE}) between neighbouring cell assemblies.

To account for the reduced inhibitory firing in the atomoxetine neural network, the extent to which we increased the w_{IE} weights between neighbouring cell assemblies was lower in the atomoxetine compared to the placebo neural network. Specifically, the w_{IE} connections between neighbouring cell assemblies were increased to $w_{IE} = 0.9 \pm 0.05$ nS for the placebo neural network, and $w_{IE} = 0.7 \pm 0.05$ nS for the atomoxetine neural network. Notably, for the atomoxetine simulation, this increase in w_{IE} was the minimum strengthening required to prevent runaway excitation following w_{EE} potentiation between the neighbouring cell

assemblies. Before and after these manual changes in synaptic weight, all w_{EE} and w_{IE} synapses remained plastic and were subject to the codependent plasticity rule, as above.

Therefore, despite w_{EE} being strengthened to the same values in both the atomoxetine and placebo neural networks, graded w_{EE} potentiation between cell assemblies was observed in the atomoxetine but not the placebo neural networks (Figure 4C, Supplementary Figure 4C). This observed difference can be attributed to differential plasticity operating on the w_{EE} and w_{IE} synapses in response to external drive applied to assembly '1', due to the reduced inhibitory firing in the atomoxetine neural network.

We have now summarised the above in the revised *Methods* and *Results* sections.